# BREAKING DOWN AND BUILDING UP: MIXTURE OF SKILL-BASED VISION-AND-LANGUAGE NAVIGATION AGENTS

## ABSTRACT

Vision-and-Language Navigation (VLN) poses significant challenges for agents to interpret natural language instructions and navigate complex 3D environments. While recent progress has been driven by large-scale pre-training and data augmentation, current methods still struggle to generalize to unseen scenarios, particularly when complex spatial and temporal reasoning is required. In this work, we propose SkillNav, a modular framework that introduces structured, skill-based reasoning into Transformer-based VLN agents. Our method decomposes navigation into a set of interpretable atomic skills (*e.g.*, Vertical Movement, Area and Region Identification, Stop and Pause), each handled by a specialized agent. To support targeted skill training without manual data annotation, we construct a synthetic dataset pipeline that generates diverse, linguistically natural, skill-specific instruction-trajectory pairs. We then introduce a novel training-free Vision-Language Model (VLM)-based router, which dynamically selects the most suitable agent at each time step by aligning sub-goals with visual observations and historical actions. SkillNav obtains competitive results on commonly-used benchmarks, and establishes state-of-the-art generalization to the GSA-R2R, a benchmark with novel instruction styles and unseen environments.

## 1 INTRODUCTION

Vision-and-Language Navigation (VLN) (Anderson et al., 2018; Zhang et al., 2024c) is a critical sub-field of embodied AI that integrates natural language understanding, visual perception, and sequential decision-making to allow autonomous agents to navigate and interact within visual environments. With the rise of foundation models (Zhou et al., 2024a; Xiao & Zhu, 2025; Li et al., 2024; Zhang et al., 2024a), VLN has seen notable progress in multimodal grounding and generalization.

Despite recent advances, a key challenge in VLN lies in enabling agents to generalize reliably and interact with unseen environments and novel instructions. Previous approaches have enhanced VLN agents' generalization ability through extensive training on large-scale synthetic instruction-trajectory pairs across varied environments (Hao et al., 2020; Chen et al., 2022a; Wang et al., 2023; 2024c). While data-driven methods improve VLN agents' generalization, their main limitation is reliance on black-box, end-to-end models (Anderson et al., 2018; Hong et al., 2021) that

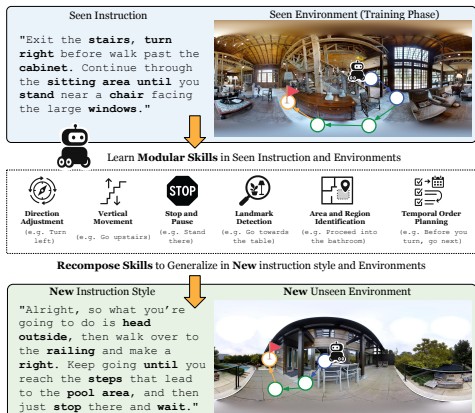

Figure 1: SkillNav decomposes complex navigation instructions into atomic skills, which can be flexibly recomposed to address new environments.

tend to memorize training examples. This restricts their effectiveness in unobserved scenarios requiring deeper compositional reasoning, such as understanding diverse instructions, temporal relationships, or complex landmarks, and generalizing across a wide range of visual environments.

Beyond data-driven approaches, recent work has explored zero-shot approaches leveraging Large Language Models (LLMs) for VLN tasks to improve generalization ability (Zhou et al., 2023; Long et al., 2024; Chen et al., 2024; Zhang et al., 2025a). Although zero-shot LLM-based agents show relatively stable performance across seen and unseen environments, they still considerably lag behind fine-tuned VLN models. Specifically, we observe a significant performance gap (approximately 36% in Success Rate), primarily arising from intrinsic limitations of LLMs, including their insufficient grounding in embodied environments and imprecise alignment of linguistic instructions with specific navigational actions. This gap highlights the urgent need for methods that combine the broad generalization and compositional reasoning capabilities of LLMs with the domain-specific adaptability of fine-tuning strategies.

To address these limitations, we propose **SkillNav**, a modular VLN framework that decomposes navigation learning into individual and reusable skills, enabling flexible re-composition and enhanced generalization in new environments (as shown in Figure 1). Unlike prior methods that treat instruction execution as an end-to-end mapping from instructions directly to actions, SkillNav explicitly captures the compositional nature of navigation tasks. Furthermore, we introduce a novel Vision-Language Model (VLM)-based router that leverages multi-modal reasoning to dynamically select the most appropriate skill at each navigation step, conditioned on the current sub-instruction, visual observation, and historical actions. SkillNav not only improves interpretability by making the decision-making processes more transparent but also facilitates robust adaptation to diverse instructions and unseen visual environments.

Specifically, we build on previous research (Wang et al., 2024b), and identify a set of atomic skills required for effectively completing the VLN task. For each skill, we construct a dataset containing relevant instructions paired with corresponding visual observations, and fine-tune a dedicated agent on top of a strong VLN backbone. This process yields five specialized skill agents, each proficient in its designated capability. After obtaining these agents, we then integrate them into a unified framework to perform complex navigation tasks. Moreover, we introduce a temporal reordering module to generate chronologically ordered sub-goals, facilitating effective temporal reasoning during skill selection. Finally, we integrate a VLM-based router that dynamically identifies the next relevant sub-goal and selects the most suitable skill-based agent to execute the corresponding navigation action.

SkillNav attains a strong performance on the Room-to-Room (R2R) benchmark (Anderson et al., 2018), and achieves state-of-the-art (SOTA) generalization to the GSA-R2R benchmark (Hong et al., 2025) which introduces novel instructions and diverse visual environments, including both unseen residential and non-residential settings. Additionally, we evaluate individual skill-based agents using NavNuances (Wang et al., 2024b), a dataset specifically designed for fine-grained skill evaluation. We provide comprehensive ablation studies and qualitative analysis to thoroughly assess the effectiveness of each component within our framework and justify our router design choices. Our contributions are summarized as follows:

1. We propose **SkillNav**, a modular framework that explicitly decomposes the navigation task into atomic, reusable skills, then recomposes them for execution, leveraging the specialization of fine-tuned VLN architectures together with the generalization capability of VLMs. This design significantly enhances generalization to novel instructions and visual environments.
2. We construct a synthetic dataset pipeline that enables skill-specific supervision without human annotation, producing diverse and linguistically natural data.
3. We demonstrate SOTA generalization on the challenging GSA-R2R dataset and provide a comprehensive analysis with ablation studies.

## 2 RELATED WORK

**Vision-and-Language Navigation Models.** A wide range of methods have been proposed for addressing VLN tasks. These methods have evolved from early LSTM-based architectures (Anderson et al., 2018; Tan et al., 2019) to Transformer-based models (Chen et al., 2021; 2022b; An et al., 2023) and, most recently, to Large Language Model (LLM)-based agents (Zhou et al., 2023; Chen et al., 2024; Lin et al., 2024; Zhou et al., 2024b; Zheng et al., 2024; Zhang et al., 2025b). A critical challenge in VLN research is enhancing the generalization capability of agents, allowing them to navigate

effectively in unfamiliar environments and handle novel instructions. To enhance generalization, most existing methods utilize data-driven augmentation strategies, focusing either on augmenting visual observations (Li et al., 2022; Liu et al., 2021; Li & Bansal, 2023) or synthesizing additional navigation instructions (Wang et al., 2023; 2024c; Hao et al., 2020; Zhang & Kordjamshidi, 2023; Zhang et al., 2024b). However, a fundamental limitation of purely data-driven augmentation approaches lies in their reliance on end-to-end training paradigms. Such monolithic models often memorize training examples rather than genuinely generalize, failing to fundamentally address the compositional reasoning required in novel or unseen scenarios. More recently, some approaches (Zhou et al., 2023; Chen et al., 2024; Long et al., 2024; Zhang et al., 2025a) have explored zero-shot navigation by heavily depending on the general reasoning capabilities of LLMs without explicit training on task-specific datasets. However, their effectiveness remains constrained by the LLMs' inherent lack of detailed spatial understanding and precise grounding in real-world action execution. In contrast, we propose SkillNav, a modular framework that explicitly decomposes VLN tasks into reusable navigation skills. Each skill is individually fine-tuned for precise spatial grounding, while high-level reasoning and flexible skill composition leverage LLMs and VLMs, significantly improving generalization to unseen environments and varied instructions.

**Skill-based MoE Systems.** Mixture-of-Experts (MoE) models traditionally operate at the parameter level, distributing input across multiple expert networks to improve capacity and efficiency (Jacobs et al., 1991; Jordan & Jacobs, 1994; Yuksel et al., 2012). Sparsely activated MoEs (Shazeer et al., 2017; Lepikhin et al., 2021; Zhang et al., 2021; Zuo et al., 2022) further scale this idea by routing each input to a small subset of experts, making it possible to train trillion-parameter models while controlling inference cost. More recently, large language models have begun to employ skill-based MoEs at the module or LLM level, where different LLMs are specialized through fine-tuning or task profiling (Riquelme et al., 2021; Wang et al., 2024a; Dai et al., 2024; Jiang et al., 2024; Xue et al., 2024; Chen et al., 2025; Zhou et al., 2024c; Yu et al., 2025), and expert selection is performed via prompting or routing mechanisms based on task semantics. While these skill-based MoE methods focus on video understanding (Yu et al., 2025) and visual or textual question-answering (Chen et al., 2025), they largely overlook embodied tasks such as VLN. Although a recent model, SAME (Zhou et al., 2024c), introduces a state-adaptive MoE framework for VLN, this approach lacks explicit skill representations and independent spatial grounding, limiting its interpretability and extensibility. In contrast, our framework explicitly defines skill-based MoE agents for VLN tasks, employing specialized skills to significantly enhance generalization, interpretability, and extensibility.

## 3 PRELIMINARIES

In the VLN problem setting, an agent navigates through an environment by following a natural language instruction $I$ to reach a specified target location. The environment is discretized into a connectivity graph $\mathcal{G} = (V, E)$, where $V$ denotes a non-empty set of navigable nodes, and $E$ is a set of undirected connectivity edges. At each time step $t$, the agent located at viewpoint $v_t$ receives a panorama represented by $n$ images, denoted as $D_t = \{o_i\}_{i=0}^n$. The agent is aware of a subset of views $O_t \subseteq D_t$ heading towards its navigable neighboring nodes $\mathcal{N}(v_t)$. The local action space $A_t$ contains navigating to node $v \in \mathcal{N}(v_t)$ or stopping at current node $v_t$.

In this work, we leverage DUET (Chen et al., 2022b) as our base VLN agent. It is a dual-scale graph transformer solution that fuses the topological map with local observations for decision-making. We formulate it as

$$a_t^* = \pi(I, O_t, M_t). \tag{1}$$

where $M_t \subseteq \mathcal{G}$ denotes the online constructed topological map observed after $t$ steps of navigation, and $a_t^* \in A_t$ is the predicted action.

## 4 METHODOLOGY

We propose a framework, **SkillNav**, for VLN that coordinates a set of atomic skill-based agents to solve navigation tasks. SkillNav enhances generalization by treating navigation as a composition of atomic skills rather than a direct language-to-action mapping. This design mirrors how humans transfer sub-skills across unfamiliar situations, preventing overfitting to specific trajectories and enabling systematic reuse of skills across environments and instruction styles. As shown in Figure 2, the

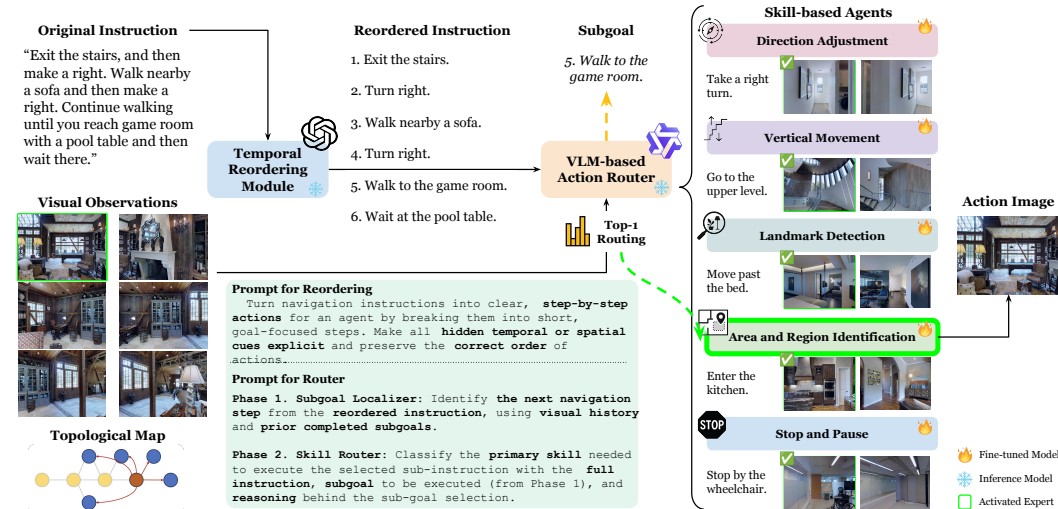

Figure 2: SkillNav Architecture. SkillNav takes visual observations, original instructions and the topological map as input. A temporal reordering module first leverages an LLM to reorder instructions into structured action goals. Subsequently, a VLM-based action router localizes the current focused sub-goal and dynamically selects the most suitable skill-based agent. For each skill, we construct specialized instruction-visual observation datasets for targeted skill learning.

framework comprises three components: a temporal reordering module for instruction decomposition, a VLM-based router for skill selection, and a set of skill-specific agents. Each agent is built upon the DUET architecture and trained with tailored synthetic data to make skill-conditioned decisions. This section introduces the proposed skill taxonomy, skill-specific synthetic dataset construction, and reasoning framework for acquiring these modular skills.

## 4.1 SKILL TAXONOMY

We use the defined skills in NavNuances (Wang et al., 2024b) that appear to be essential for building a robust VLN agent. NavNuances provides skill categories and creates a diagnostic dataset to analyze models' errors. However, it does not provide solutions for improving the agent skills. In this work, we extend the initially proposed skill categories and provide solutions for acquiring them by the skill-based agents. We adopt four frequently observed atomic skills from NavNuances, **Direction Adjustment**, **Vertical Movement**, **Landmark Detection**, and **Area and Region Identification**. Moreover, we find persistent challenges in temporal reasoning and stop criteria. Errors in temporal reasoning often disrupt the correct order of subgoal execution. Critical stop decisions are sometimes made too early or too late, reducing navigation success. To address these issues, we extend the skill taxonomy with two additional skills: **Stop and Pause** and **Temporal Order Planning**. In the following, we elaborate on these two new skills and their roles in navigation.

**Stop and Pause** captures the agent's ability to dynamically control motion termination and temporary halting in response to visual or linguistic cues. This includes recognizing explicit stop commands (*e.g.*, "Stop at the doorway") or context-sensitive halts triggered by landmarks or obstacles (*e.g.*, "Pause when you see the red sign"). The stop and pause skill emphasizes precise temporal-spatial control to ensure safe, context-aware navigation.

**Temporal Order Planning** reflects the agent's capability to reason over the sequence and structure of subgoals. This includes understanding conditional immediacy (*e.g.*, "Once you enter the hallway, turn left"), maintaining actions for a bounded duration (*e.g.*, "Keep walking until you see the staircase"), executing forward sequential steps (*e.g.*, "Go forward, then turn right, and finally stop"), and handling backward references to prior states (*e.g.*, "Before turning, make sure you're at the hallway entrance"). Effective temporal order planning involves temporal relations that guide both when and how atomic skills should be executed.

To quantify the presence and frequency of these skills in R2R (Anderson et al., 2018), we perform a keyword-based analysis of the navigation instructions as shown in Figure 4 in Appendix A. Each

instruction is scanned for a curated set of indicative keywords, compiled for each skill category based on linguistic patterns observed in prior datasets and real-world navigation discourse. For instance, terms like "wait" or "stay" are used to detect Stop and Pause, while words such as "stairs" or "elevator" signal Vertical Movement. An instruction can be counted for multiple skills if it exhibits multiple relevant keywords.

## 4.2 Skill-Specific Data Synthesis and Agent Training

To enable the training of skill-specialized agents, we construct a set of synthetic datasets in which each trajectory–instruction pair is specifically designed to emphasize a single navigation skill.

We begin with a random starting node in the Matterport3D (Chang et al., 2017) environment and sample diverse navigation paths through graph traversal. For each skill, we define filtering heuristics to select trajectories where this skill is the primary factor for successful navigation. For instance, we emphasize frequent orientation changes or non-trivial turning sequences for the Direction Adjustment category. We explain detailed primary factors of skill-based trajectory generation in Appendix A. Each selected trajectory consists of a sequence of panoramic observations.

Table 1: Statistics of skill-specific synthetic datasets and existing VLN training datasets.

| Dataset | # Instr | # Vocab | Instr Len |
|---|---|---|---|
| R2R | 14,039 | 4,597 | 26.28 |
| GSA-R2R | 4,675 | 2,797 | 26.06 |
| Temporal | 2,000 | 1,653 | 56.60 |
| Direction | 450 | 707 | 26.78 |
| Vertical | 450 | 705 | 26.23 |
| Stop | 450 | 774 | 27.03 |
| Landmark | 450 | 1,025 | 27.62 |
| Region | 450 | 971 | 27.50 |

vations. Besides, we constrain trajectory length to 4–7 steps to keep the difficulty and temporal context comparable to human-annotated VLN data. The analysis of path length constraints used during trajectory generation to ensure temporal consistency and alignment with existing VLN datasets are provided in Figure 5 in Appendix B.

To generate skill-focused instruction, we feed the observation sequence of each candidate trajectory into GPT-4o (OpenAI, 2024) with a structured prompt. We design the prompts such that the generated instructions preserve the general linguistic quality of real VLN datasets, including comparable sentence length, vocabulary diversity, and fluency, while emphasizing the content toward the targeted skill. This is achieved by providing GPT-4o with explicit skill-focused cues during generation, encouraging, for example, frequent references to orientation change for the Direction Adjustment skill or strong emphasis on landmark description for the Landmark Detection skill. For each skill, we synthesize $N$ such trajectory–instruction pairs, forming six separate datasets. A summary of dataset statistics is provided in Table 1.

The training of each skill-based agent is conducted in two stages. In the first stage, we fine-tune the pre-trained DUET model using the original R2R training dataset, the ScaleVLN augmentation data (Wang et al., 2023), and our Temporal Synthetic dataset to obtain a strong, skill-agnostic backbone. We provide the analysis of the effectiveness of the Temporal Order Planning agent in Appendix C. In the second stage, this backbone is further fine-tuned on a skill-specific synthetic dataset to obtain a specialized agent in the targeted skill. Following this process, we obtain five specialized skill-based agents: the Direction Adjustment agent ($\pi_{da}$), Vertical Movement agent ($\pi_{vm}$), Stop and Pause agent ($\pi_{sp}$), Landmark Detection agent ($\pi_{ld}$), and Area and Region Identification agent ($\pi_{ar}$). We denotes the predefined set of five skill-based agents as $\mathcal{S} = \{\pi_{da}, \pi_{vm}, \pi_{sp}, \pi_{ld}, \pi_{ar}\}$.

## 4.3 SkillNav Framework

After training specialized agents for different navigation skills, we build our SkillNav framework. SkillNav first employs a temporal reordering module to generate chronologically ordered execution plans. Then, we introduce a VLM-based action router to accurately identify the current subgoal and dynamically select the corresponding skill-based agent to choose the appropriate action.

### 4.3.1 Temporal Reordering Module

The Temporal Reordering Module only takes the original natural language instruction as input. It applies the instruction reordering prompt to turn navigation instructions into a list of subgoals $I_{reorder}$.

Table 2: Performance comparison on R2R and GSA-R2R benchmarks. † indicates large-scale data augmentation. SRDF performs best on R2R due to extensive pretraining on data that mimics R2R-style instructions; however, it struggles to generalize effectively to the GSA-R2R dataset.

| Methods | # | R2R | | | | | | | | GSA-R2R | | | | | |
| --- | --- | --- | --- | --- | --- | --- | --- | --- | --- | --- | --- | --- | --- | --- | --- |
| | | Val-Unseen | | | | Test-Unseen | | | | Test-R-Basic | | Test-N-Basic | | Test-N-Scene | |
| | | NE↓ | OSR↑ | SR↑ | SPL↑ | NE↓ | OSR↑ | SR↑ | SPL↑ | SR↑ | SPL↑ | SR↑ | SPL↑ | SR↑ | SPL↑ |
| *LLM-based VLN* | | | | | | | | | | | | | | | |
| MapGPT (GPT4v) (Chen et al., 2024) | 1 | 5.63 | 58 | 44 | 35 | – | – | – | – | 34 | 30 | 25 | 23 | 25 | 23 |
| NavCoT (LLaMA2) (Lin et al., 2024) | 2 | 6.26 | 42 | 34 | 29 | – | – | – | – | 37 | 35 | 29 | 26 | 29 | 26 |
| NavGPT-2 (FlanT5-5B) (Zhou et al., 2024b) | 3 | 3.13 | 81 | 72 | 61 | 3.33 | 80 | 72 | 60 | 58 | 45 | 48 | 35 | **57** | 43 |
| NaviLLM (Vicuna-7B) (Zheng et al., 2024) | 4 | 3.51 | – | 67 | 59 | 3.71 | – | 68 | 60 | – | – | – | – | – | – |
| *Supervised VLN* | | | | | | | | | | | | | | | |
| HAMT (Chen et al., 2021) | 5 | 2.29 | – | 66 | 61 | 3.93 | 72 | 65 | 60 | 48 | 44 | 42 | 38 | 34 | 30 |
| DUET (Chen et al., 2022b) | 6 | 3.31 | 81 | 72 | 60 | 3.65 | 76 | 69 | 59 | 58 | 47 | 48 | 37 | 40 | 30 |
| BEVBERT (An et al., 2023) | 7 | 2.81 | 84 | 75 | 64 | 3.13 | 81 | 73 | 62 | 58 | 45 | 46 | 35 | 39 | 27 |
| GR-DUET (Hong et al., 2025) † | 8 | – | – | – | – | – | – | – | – | 69 | 64 | 57 | 52 | 48 | 43 |
| ScaleVLN (Wang et al., 2023) † | 9 | 2.34 | 87 | 79 | 70 | 2.73 | 84 | 77 | 68 | 78 | 67 | 69 | 57 | 55 | 43 |
| SRDF (Wang et al., 2024c) † | 10 | 1.83 | **89** | **84** | **78** | 1.88 | **88** | **84** | **77** | 71 | 63 | 59 | 49 | 52 | 43 |
| *Mixture of Skill-based VLN* | | | | | | | | | | | | | | | |
| SAME† (Zhou et al., 2024c) | 11 | 2.73 | – | 76 | 66 | 3.03 | – | 74 | 64 | – | – | – | – | – | – |
| SkillNav† (ours) | 12 | 1.97 | **89** | 83 | 77 | 2.53 | 83 | 78 | 70 | 79 | 69 | 72 | 61 | 57 | 48 |

It follows the four temporal relations described in the Temporal Order Planning skill in Section 4.1, making implicit temporal details explicit and ensuring the correct subgoal execution order. This procedure is formulated as

$$I_{\text{reorder}} = \text{LLM}_{\text{TemporalReorder}}(I). \tag{2}$$

### 4.3.2 VLM-BASED ACTION ROUTER

To coordinate skill-based agents during navigation, we introduce an Action Router that dynamically selects the most suitable agent at each time step. Inspired by LLM-based planning systems such as LLM-Planner (Song et al., 2023), Mic (Qiao et al., 2023), and A2Nav (Chen et al., 2023), our router leverages a large VLM model (e.g., GPT-4o (OpenAI, 2024), Qwen2.5-VL-7B-Instruct (Bai et al., 2025)) in a zero-shot in-context fashion. We structure the routing process into two distinct reasoning phases:

**Phase 1: Subgoal Localizer.** Given the reordered subgoals $I_{\text{reorder}} = [p_1, p_2, \ldots, p_m]$, observed history $H_{t-1}$, and the sequence of previously executed subgoals $G_{t-1} = [p_1^*, \ldots, p_{t-1}^*]$, the model identifies the next subgoal $p_t^*$ to be executed for the current time step $t$ and outputs the corresponding reasoning trace $r_t$, later used by the router for decision verification. The output can be formalized as:

$$p_t^*, r_t = \texttt{Localize}(I_{\text{reorder}}, H_{t-1}, G_{t-1}). \tag{3}$$

The sequence of executed subgoals is then updated as:

$$G_t = G_{t-1} \parallel p_t^*. \tag{4}$$

**Phase 2: Skill Router.** At time step $t$, the skill router determines which skill-based agent $\pi_t^* \in \mathcal{S}$ is most appropriate for executing the selected subgoal $p_t^*$. Besides, it receives the original instruction $I$ as a part of the input context to capture additional linguistic cues such as verbs and spatial references. It also uses the reasoning trace $r_t$ from Phase 1 to enhance its understanding of the current subgoal. At each step, exactly one skill is selected, formulated as

$$\pi_t^* = \arg\max_{\pi \in \mathcal{S}} \texttt{Router}(I, p_t^*, r_t). \tag{5}$$

Once the appropriate skill-based agent is selected, it is invoked by the following Equation 1 to predict the navigation action at time step $t$:

$$a_t^* = \pi_t^*(I, O_t, M_t). \tag{6}$$

Our router enables modular skill execution by integrating natural language, visual inputs, and observed history, using the Temporal Reordering LLM to bridge instructions with actionable skill modules.

## 5 EXPERIMENTS

**Evaluation Datasets.** We primarily use the Room-to-Room (R2R) dataset (Anderson et al., 2018), especially the unseen split of validation (Val Unseen) and test (Test Unseen) splits. R2R is a commonly-used benchmark in VLN consisting of panoramic RGB-D scans from the Matterport3D (Chang et al.,

Table 3: Evaluation of each skill-based agent on the NavNuances benchmark across four skill categories: Direction Change (DC), Vertical Movement (VM), Landmark Recognition (LR), and Room Recognition (RR). Following the NavNuances, evaluation metrics differ across skill subsets: DC and LR are reported only with SR, VM includes SR/OSR/SPL, and RR provides SR/OSR. We retain this heterogeneous metric design to ensure comparability with prior work. Ident.: Identification.

| Methods | | DC | VM | | | LR | RR | |
|---|---|---|---|---|---|---|---|---|
| | | SR | SR | OSR | SPL | SR | SR | OSR |
| **VLN Agents** | ScaleVLN (Wang et al., 2023) | 68.39 | 81.76 | 88.82 | 76.34 | 28.32 | 82.91 | 95.27 |
| | SRDF (Wang et al., 2024c) | 59.93 | 82.94 | **91.18** | 80.98 | 26.28 | 77.09 | 94.55 |
| **Skill-based Agents** | Direction Adjustment | **70.81** | 81.76 | **91.18** | 76.28 | 31.39 | 81.82 | 94.91 |
| | Vertical Movement | 70.68 | **87.65** | 89.41 | **83.83** | 30.22 | 82.18 | 96.00 |
| | Landmark Detection | 70.29 | 82.35 | 85.29 | 78.94 | **31.53** | 83.64 | **97.09** |
| | Area and Region Ident. | 67.53 | 84.12 | 88.82 | 80.49 | 29.20 | **85.09** | 96.36 |
| | Stop and Pause | 68.91 | 84.71 | 87.06 | 80.67 | 29.78 | 83.64 | **97.09** |

2017) simulator and providing crowd-sourced instructions paired with navigation paths. Moreover, we evaluate the generalization ability of SkillNav on GSA-R2R (Hong et al., 2025) which includes residential (R) and non-residential (N) scenes (*e.g.*, shops, restaurants, and museums) from Habitat-Matterport3D (Ramakrishnan et al., 2021), and diverse instruction styles with role-specific dialogues (*e.g.*, travel guides (Scene) beyond the basic style of R2R (Basic).

**Evaluation Metrics.** We use the standard metrics to evaluate the navigation performance (Anderson et al., 2018; Zhao et al., 2023): (1) Navigation Error (NE): the distance between the stop location and the target; (2) Oracle Success Rate (OSR): the agent ever gets close enough to the goal at any point along its trajectory, regardless of where it decides to stop; (3) Success Rate (SR): the ratio of agents stopping within 3 meters of the target; (4) Success rate weighted by Path Length (SPL): measure navigation efficiency by weighting the success rate with the ratio between the shortest path length and the agent's actual path length, penalizing unnecessarily long trajectories.

**Implementation Details.** We utilize CLIP-B/16 (Radford et al., 2021) as the visual backbone and BERT-base-uncased (Devlin et al., 2018) as the language backbone within our DUET-based skill agents. During the skill training, we fine-tune the DUET pre-trained model with Temporal Order synthetic data, ScaleVLN augmentation data, and R2R Train data for $50,000$ iterations using a batch size of 32 and a learning rate of $5 \times 10^{-5}$ on 1 NVIDIA A6000 GPU with the random seed 0. The best finetuned Temporal DUET model is selected based on the SPL performance on the R2R Validation Unseen dataset. Based on the Temporal DUET, we employ the second round fine-tuning with atomic skill synthetic data for $30,000$ iterations with a batch size of 16 on the same GPU. In our SkillNav LLM-based architecture, we adopt GPT-4o (OpenAI, 2024) as the Temporal Reordering module due to its superior instruction-following capabilities and employ Qwen2.5-VL-7B-Instruct (Bai et al., 2025) as the action router because of its strong multi-modal alignment and reasoning abilities. All inferences with the action router are performed using in-context prompting.

## 5.1 MAIN RESULTS

As shown in Table 2, SkillNav achieves strong overall performance across both R2R datasets and demonstrates robust generalization on GSA-R2R, outperforming most fine-tuned and LLM-based agents. On the R2R unseen environments, SkillNav (Method #12) achieves 83% SR and 77% SPL, ranking second highest after SRDF (Method #10). While SRDF achieves the highest performance on R2R Test-Unseen, this can be largely attributed to its pretraining on large-scale data that closely follows R2R-style instruction patterns. However, this reliance weakens its generalization ability, leading to a 13% and 5% SR drop on GSA-R2R Test-N-Basic and Test-N-Scene, respectively. SRDF requires additional tuning to remain competitive when transferred to new environments or novel instruction styles. In contrast, SkillNav is trained only on R2R and synthetic skill-specific data, yet achieves strong cross-dataset generalization without any retraining. Additionally, SkillNav also demonstrates SOTA generalization performance in GSA-R2R, ranking 1st in SPL across all GSA-R2R splits and demonstrating its ability to predict more efficient and precise navigation trajectories. Notably, on Test-N-Scene, which combines non-residential environments with more complex and role-specific instructions, SkillNav matches the best SR tied with NavGPT-2 (Method #3), while significantly outperforming it in SPL. NavGPT-2 benefits from fine-tuning on FlanT5-XXL (Chung

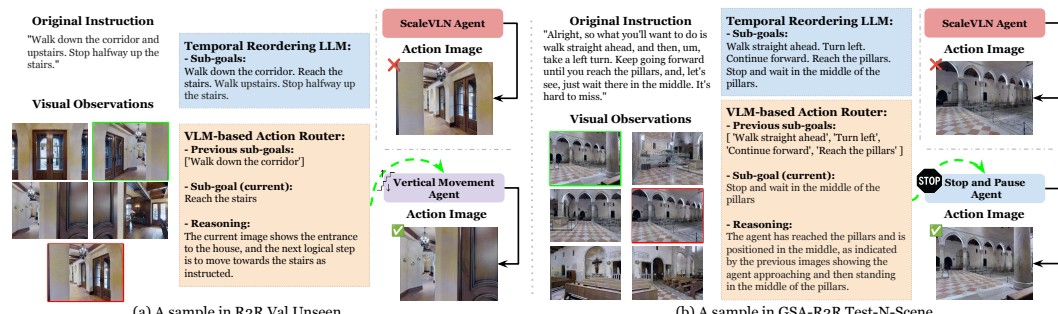

(a) A sample in R2R Val Unseen
(b) A sample in GSA-R2R Test-N-Scene

Figure 3: Qualitative examples of routing and navigation results. These examples include cases where the instruction is temporally complex, colloquial, or spatially ambiguous.

Table 4: Ablation results on GSA-R2R across residential (R) and non-residential (N) scenarios with varying instruction styles (Basic and Scene). Reorder: ✗ = LLM-guided Temporal Reordering disabled, ✔ = enabled. Router: Random = randomly select skill-based agents without utilizing action router; Qwen = Qwen2.5-VL-7B-Instruct; GLM = GLM-4.1V-9B-Thinking.

| Reorder | Router | # | Test-R-Basic | | Test-N-Basic | | Test-N-Scene | |
|---|---|---|---|---|---|---|---|---|
| | | | SR | SPL | SR | SPL | SR | SPL |
| ✗ | Random | 1 | 78.39 | 67.46 | 70.93 | 59.71 | 54.61 | 43.17 |
| ✗ | Qwen | 2 | 78.42 | 67.80 | 71.01 | 59.62 | 55.46 | 45.43 |
| ✔ | GLM | 3 | 78.60 | 67.93 | 71.13 | 59.73 | **56.80** | 46.51 |
| ✔ | Qwen | 4 | **78.83** | **68.88** | **71.58** | **61.34** | 56.66 | **47.96** |

et al., 2022), which likely enhances its ability to interpret stylized instructions. However, its lower SPL reveals inefficiencies in path planning and execution. While LLMs can help parse diverse instructions, they often introduce noise or lose critical spatial details when translating, limiting their effectiveness in downstream navigation tasks. This highlights the need for tightly integrated skill reasoning and grounded visual understanding, beyond language interpretation alone.

## 5.2 ABLATION STUDY

**Skill Evaluation.** To further probe the capabilities of our skill-based agents, we have a fine-grained evaluation on the NavNuances, which categorizes navigation instructions into four atomic skills: (1) Direction Change (DC), (2) Vertical Movement (VM), (3) Landmark Recognition (LR), and (4) Region Recognition (RR). These subsets isolate specific reasoning capabilities and allow us to assess each agent's specialization. As shown in Table 3, each skill-based agent in SkillNav excels in its corresponding category. The Vertical Movement agent achieves the highest SR (87.65%) and SPL (83.83%) on VM, while the Direction Adjustment agent leads in DC with an SR of 70.81%. The Landmark Detection agent performs best in LR with 31.53% SR, and the Area and Region Identification agent reaches 85.09% SR on RR. We report the effectiveness of the Stop and Pause agent in Appendix D. These results validate our skill-based training and data augmentation strategy, confirming that targeted supervision fosters functional specialization that outperforms generalist VLN baselines in isolated skill settings.

**Temporal Reordering Module.** We conduct an ablation study to evaluate SkillNav's two key components: the LLM-guided Temporal Reordering module and the VLM-based action router. The results, shown in Table 4, are reported across GSA-R2R splits, covering both residential (R) and non-residential (N) environments with varying instruction styles. First, we evaluate the effectiveness of the temporal reordering module. As shown in rows #2 and #4, when using the same router (Qwen2.5-VL-7B-Instruct), incorporating the reordering module consistently improves performance across all benchmarks. Notably, in Test-N-Basic, SPL increases +1.72%, demonstrating that temporally structured subgoals offer clearer guidance for effective skill selection.

**Action Router.** To evaluate the effectiveness of our action router, we compare the performance of randomly selected skills without a router (row #1) against our proposed Qwen router. The observed improvements in both SR and SPL metrics clearly indicate the router's effectiveness: specifically,

Test-N-Scene SR increases from $54.61\%$ to $55.46\%$, and SPL rises notably from $43.17\%$ to $45.43\%$. These results confirm that our VLM-based router effectively selects appropriate skills even in the absence of temporal structuring. We further examine the significance of router selection by comparing rows #3 and #4, where the instruction reordering is fixed, and only the router model varies. Qwen2.5-VL-7B-Instruct consistently achieves superior SPL across all splits, particularly notable in Test-N-Scene ($47.96\%$ vs. $46.51\%$), underscoring its enhanced visual grounding capabilities compared to GLM-4.1V-9B-Thinking (Team et al., 2025). This emphasizes that high-quality vision-language representations are essential for effective skill routing.

## 5.3 EFFICIENCY ANALYSIS

**Training Cost.** Fine-tuning five skills on the Temporal Order Planning agent with R2R and synthetic skill-specific datasets requires approximately $3,329$ minutes ($\sim 55.5$ hours) in total. For comparison, SRDF training on R2R with larger data augmentation takes $2,521$ minutes ($\sim 42$ hours), suggesting that SkillNav's skill-based training introduces a relatively higher training cost. However, this represents a one-time training investment; unlike prior supervised VLN models that require repeated retraining to adapt to new environments or instruction styles, SkillNav achieves strong generalization across datasets without additional retraining.

**Inference Cost.** We provide inference time and throughput comparison in Table 5. SkillNav introduces overhead due to its Temporal Reordering LLM and VLM-based action router, reaching $0.49$ throughput on Test-N-Basic of GSA-R2R, which is roughly $50\times$ slower than ScaleVLN but still nearly $20\times$ faster than MapGPT. The Random variant, despite sharing the DUET as the backbone and selecting only one DUET for action prediction, is $4.3\times$ slower than ScaleVLN due to the per-observation skill selection overhead that prevents batch inference. Overall, while SkillNav is less efficient than supervised models, it achieves a better efficiency-generalization trade-off. Also, it advances both efficiency and generalization compared to LLM-based VLN agents.

Table 5: Runtime and throughput of baselines and SkillNav. Numbers are wall-clock runtime in seconds. Random = randomly select skill-based agents without utilizing the action router.

| Method | Split | Runtime (s) | Inferences/s |
|---|---|---|---|
| *Supervised VLN* | | | |
| ScaleVLN | Test-R-Basic | 513.8 | **28.03** |
| | Test-N-Basic | 342.7 | **26.26** |
| *LLM-based VLN* | | | |
| MapGPT | Test-R-Basic | $\sim 597,000$ | 0.02 |
| | Test-N-Basic | $\sim 373,000$ | 0.02 |
| *Our Mixture of Skill-based VLN* | | | |
| Random (ours) | Test-R-Basic | 2,223.4 | 6.48 |
| | Test-N-Basic | 1,507.9 | 5.97 |
| SkillNav (ours) | Test-R-Basic | $\sim 27,000$ | 0.54 |
| | Test-N-Basic | $\sim 18,360$ | 0.49 |

## 5.4 QUALITATIVE EXAMPLES

Figure 3 shows two qualitative examples highlighting SkillNav's capability to dynamically select the appropriate skill at each navigation step. These examples illustrate the effectiveness of our approach in reordering temporal action plans, accurately identifying the currently focused subgoal via the router, and subsequently selecting the correct action. Specifically, in Figure 3 (a), the router correctly reasons that the agent has reached the target pillars and decides it is time to stop, resulting in the agent appropriately choosing the stop action at the view containing the pillars. Similarly, in Figure 3 (b), the router identifies the need to move toward the stairs and accordingly selects the vertical movement skill. Overall, SkillNav successfully interprets diverse instruction styles and performs robustly across both residential and non-residential scenes.

## 6 CONCLUSION

We introduce SkillNav, a VLN agent that combines skill-based learning with VLM-based routing to dynamically select the most suitable actions based on the decision of the most relevant expert. We evaluate SkillNav on R2R to show strong navigation performance and demonstrate its generalization capabilities on the GSA-R2R dataset. While the utilization of LLM for temporal reordering and VLM for routing introduces computational overhead, SkillNav is more efficient than relying solely on LLMs or VLMs for navigation and achieves stronger performance than supervised VLN agents by exploiting both paradigms. Our framework provides a novel and interpretable approach that advances compositional reasoning and generalization for the VLN research community.

# 7  ETHICS STATEMENT

This work builds upon publicly available datasets and standard benchmarks for vision-and-language navigation, without the collection of new human subject data. All experiments are conducted in simulated environments, and no personally identifiable information is involved. We acknowledge that large language and vision-language models may encode societal biases; however, our use is limited to controlled research settings. Code and resources are released solely for academic research purposes.

# 8  REPRODUCIBILITY STATEMENT

We have taken several steps to ensure the reproducibility of our results. The datasets, evaluation metrics, and experimental protocols are described in Section 5, with complete details of data pre-processing and generation provided in Appendix A and B. Ablation studies that validate individual components are presented in Section 5.2. An anonymized version of our code and scripts is also provided in the supplementary submission.

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

APPENDIX

## A  PRIMARY FACTORS OF TRAJECTORY GENERATION

As introduced in Section Skill-Specific Data Synthesis and Agent Training in Methodology, we construct 5 skill-specific datasets and train the agents based on them. The primary factors for the construction of each skill are as follows:

**Temporal Order Planning.**  (1) A random initial move is selected. (2) Staying in the same region (*e.g.*, hallway → hallway) for the first half of the trajectory to encourage temporal continuity at first. (3) Once halfway through, the agent is allowed (and encouraged) to transition to new regions.

**Direction Adjustment.**  (1) The direction change is based on the heading degree. (2) It should be significant enough to indicate a directional shift, but not so large as to cause a reversal or double-turn behavior.

**Vertical Movement.**  (1) Only candidates with significant elevation (more than $\pm 2$) are considered, which filters out nearly flat or slight inclines/declines. (2) The candidate viewpoint must be explicitly marked as vertically relevant (*e.g.*, stairs). (3) The elevation sign determines movement type, and it must be consistent with the applied trajectory. For instance, it is impossible to go upstairs and then go downstairs in one case.

**Stop and Pause.**  (1) The stop should occur at a place with or after semantically relevant context for pausing, *e.g.*, in front of a painting, at the foot of stairs. (2) The candidate image is very similar to the previous viewpoints.

**Landmark Detection.**  (1) The viewpoint must include obvious, visually distinctive landmarks or objects (*e.g.*, sofa, desk, painting, lamp) clearly visible in the image. (2) If a landmark is to be referenced over multiple steps, it should appear persistently in successive views, allowing the agent to maintain spatial awareness relative to that object.

**Area and Region Identification.**  (1) A trajectory must include at least one region change. (2) Paths with "Error" or unrecognized regions are ignored or sanitized. (3) All horizontal region changes are isolated.

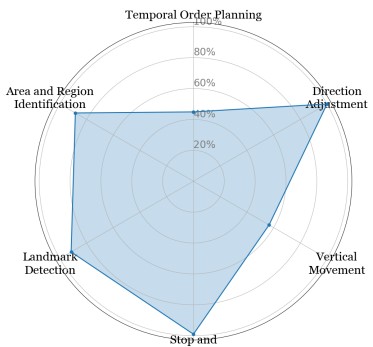

Figure 4: Distribution of instructions in the R2R dataset categorized by the proposed skill taxonomy.

## B  PATH LENGTH IN TRAJECTORY GENERATION

We constrain trajectory length to 4–7 steps to keep the difficulty and temporal context comparable to natural VLN data. Figure 5 shows the statistics of the path length. To be noted, the R2R, ScaleVLN, SRDF datasets, and our Temporal Order Planning datasets have quite less instructions with a 4-step trajectory.

## C  TEMPORAL ORDER PLANNING AGENT

As introduced earlier, the training of each skill-based agent follows a two-stage fine-tuning strategy. In the first stage, we fine-tune a pre-trained DUET model using a combination of the R2R training split, ScaleVLN augmentation data, and our proposed Temporal Synthetic dataset, resulting in a

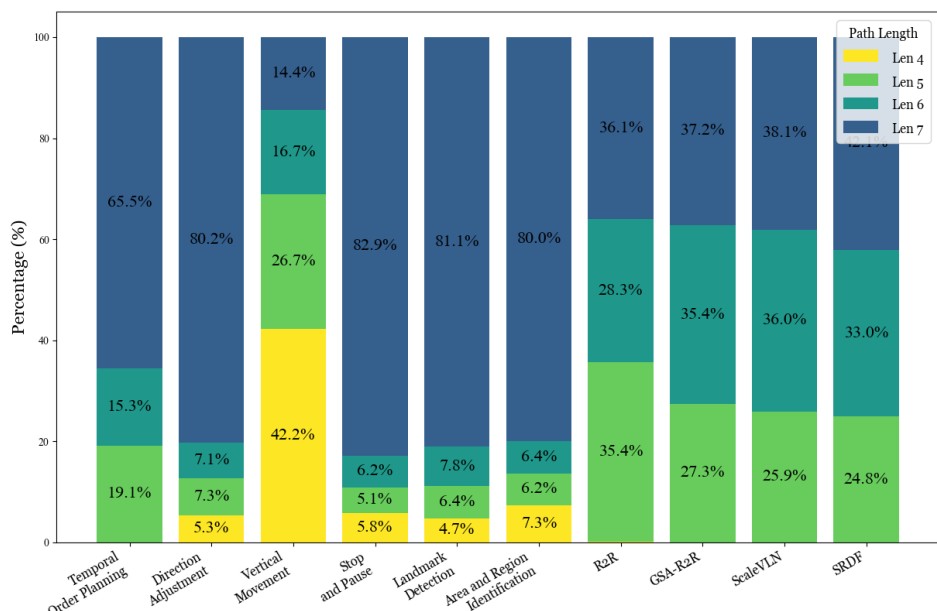

Figure 5: The statistics of the path length of our synthetic datasets compared with existing VLN datasets. The R2R, ScaleVLN, SRDF datasets, and our 6 skill-specific datasets are all for training, while only GSA-R2R is for evaluation.

strong skill-agnostic backbone. We evaluate this first-stage model on the R2R Val Unseen split across four temporal logic subsets.

Temporal Order Planning captures the agent's ability to reason over the sequence and structure of subgoals. Compared to ScaleVLN, our model demonstrates improved temporal reasoning capabilities, as detailed in Table 6. This improvement comes from enhanced **Temporal Order Planning**, which enables the agent to reason about the sequence and structure of subgoals. The Temporal Order Planning subsets include:

- **Conditional immediacy**: The agent must execute an action immediately after a specific condition is met. These instructions are typically triggered by phrases such as *once*, *as soon as*, or *upon*. (*e.g.*, "Once you enter the hallway, turn left")
- **Bounded duration**: The agent is required to maintain an action until a specific condition becomes true. These instructions use keywords such as *until* or *while*. (*e.g.*, "Keep walking until you see the staircase")
- **Forward sequential**: These instructions describe a sequence where Action B follows Action A in order. Temporal cues include *then*, *finally*, *before*, and *after*. (*e.g.*, "Go forward, then turn right, and finally stop")
- **Backward sequential**: Action B is described first but should occur only after Action A. These often use similar cues as (*e.g.*, "Before turning, make sure you're at the hallway entrance"), but the order of mention and execution differs.

Unlike low-level action chaining, temporal order planning involves higher-level temporal logic that determines when and how atomic skills should be executed in sequence. As shown in Table 6, our Temporal Synthetic Data improves navigation in failure cases where prior methods such as ScaleVLN struggle.

## D  STOP AND PAUSE AGENT

The Stop and Pause agent integrates two stopping mechanisms within the DUET framework: (1) the agent can explicitly issue a stop action at a given viewpoint; and (2) if the agent does not explicitly

Table 6: Navigation performance across 4 temporal logic instructions from R2R Val Unseen dataset. **Bold** values denote metrics that exceed the R2R Val Unseen average, while gray values indicate metrics that fall below the average. Temporal DUET is the agent fine-tuned with the Temporal Order Planning synthetic dataset in the first training stage.

| Environment | Metric | ScaleVLN | Temporal DUET |
|---|---|---|---|
| Conditional Immediacy | SR | **84.29** | **88.57** |
| | SPL | **76.29** | **82.18** |
| Bounded Duration | SR | 76.27 | **84.18** |
| | SPL | 67.45 | 74.90 |
| Forward Sequential | SR | **79.53** | **85.83** |
| | SPL | 68.92 | **76.93** |
| Backward Sequential | SR | 74.29 | **88.57** |
| | SPL | 66.97 | **81.72** |

stop when the navigation loop ends, DUET retrospectively selects the visited location with the highest stop probability and optionally appends a shortest path to reach it. Since we apply a stopping-focused data augmentation strategy that exposes the model to diverse stop-relevant cues during training, this supervision enables the agent to distinguish between the two stopping mechanisms and to learn when stopping aligns with the instruction intent and visual context. Although NavNuances does not include a dedicated stopping split, our Stop agent still outperforms generalist baselines like ScaleVLN and SRDF across all skill categories (Table 3), suggesting that effective stopping is a foundational capability that influences the success of diverse navigation behaviors.

## E EFFICIENCY ANALYSIS

All experiments in efficiency analysis in Section 5.3 run on NVIDIA A6000. For the inference cost in Table 5, the number of predictions is $14,400$ for Test-R-Basic and $9,000$ for Test-N-Basic. For fairness, MapGPT is re-implemented with Qwen2.5-VL-7B-Instruct.

## F LLM USAGE

We used LLM-based tools for polishing grammar and aiding the writing. In addition, we utilize LLM to generate synthetic instructions for skill-specific datasets as described in Section 4.2. Moreover, LLMs and VLMs serve as our temporal reordering module and action router in Section 4.3.1 and 4.3.2.

## G LIMITATIONS

First, SkillNav is evaluated only in discrete VLN simulator environments (R2R, GSA-R2R, and NavNuances), leaving open the challenge of extending to continuous or real-world robotic navigation. Second, the approach depends on synthetic, skill-specific datasets generated via prompting, which may introduce distributional biases compared to human-authored instructions. We do a human evaluation on 20 cases with action routing, and the result shows 100% accuracy. This means with high confidence, the true accuracy is at least 84% on R2R Val Unseen.

## H LLM AND VLM PROMPTS

In this section, we provide the prompts used in data construction and all components of SkillNav.

## H.1 PROMPTS FOR SKILL-SPECIFIC DATA SYNTHESIS

To generate skill-focused instruction, we feed the observation sequence of each candidate trajectory into GPT-4o with the structured prompt, in Listing 1 and Listing 2. Both of the two prompts are tailored for GPT-4o.

**Temporal Order Planning Skill Data Construction.** The detailed prompt for Temporal Order Planning Skill data construction can be seen in Listing 1.

```
You are an expert in Vision-and-Language Navigation (VLN) and Language.

<Task>
Your task is to write natural, human-like navigation instructions based on a sequence of
    visual observations from an indoor environment.

<Instruction Guidelines>
- Do not use explicit temporal markers like ``then'', ``next'', ``before'', or ``after''.
- Imply sequence using spatial or contextual phrasing instead.
- Include only essential visual cues, such as layout, furniture, doorways, or notable
    landmarks that help clarify the path.
- Avoid over-descriptive or decorative language (e.g., ``intricate stonework'', ``high
    ceiling'').
- Keep the instruction fluent, intuitive, and helpful, like someone casually guiding a
    friend through a space.
- Keep it concise and comparable in length to a temporal-based instruction.

<Visual Reasoning Process>
Analyze each frame in the visual sequence. Focus on:
- Movement across spaces
- Transitions (e.g., turns, room entries)
- Orientation shifts
- Key visible cues needed to navigate the path

<Instruction Output>
Once you've analyzed the path:
- Write a fluent, natural-sounding instruction describing the full trajectory.
- Do **not** include reasoning steps.
- Output **only** the final instruction.

<Example Chain-of-Thought>
- 1st Frame:
    - The agent is inside a narrow wooden hallway-like space.
    - The doorway directly ahead leads to a brighter area.

- 2nd Frame:
    - The agent is almost at the threshold of the doorway.
    - You can see the hallway plant and the open area outside.

- 3rd Frame:
    - The agent is now fully outside the room, looking into a wide open space.
    - There's a visible bedroom to the left, and the plant in the yellow pot is to the right
        , indicating the agent has made a hard left turn.

- 4th Frame:
    - The agent is now facing a doorway to a bedroom on the left side.
    - The bed is partially visible inside.

- 5th Frame:
    - The agent has entered the room and is facing a window.
    - The position suggests the agent took one step inside and then stopped.

---

<Trajectory Images>
``{path_images}''
```

Listing 1: Prompt used for Temporal Order Planning Skill-specific Data Synthsis

**Atomic Skills Data Construction.** The 5 atomic skills in VLN share the same prompt (in Listing 2) for their skill-specific data synthesis. .

```
You are an expert in Vision-and-Language Navigation (VLN) and Language.

<Task>
- Generate a **single** natural-language instruction that guides an agent through the scene.

<Input>
- A visual sequence (an ordered list of images)
- A specific navigation skill to emphasize

<Requirements>
- The instruction should describe what the agent does across the image sequence (e.g., move
    , climb, pause).
- Ground the instruction in **visible cues**, such as layout, objects, stairs, doorways,
    lighting, or orientation.
- Emphasize the given **target skill** (e.g., "Direction Adjustment", "Vertical Movement",
    etc.), while naturally incorporating other relevant details as needed.
- The output must be a **single sentence**, written in fluent, natural language (no lists,
    quotes, or symbols).
- Instruction length should be **20-30 words** (aim for ~25).
- Do **not** include explanations, reasoning steps, or metadata output only the instruction
     itself.

<Available Skills>
{Direction Adjustment, Vertical Movement, Stop and Pause, Landmark Detection, Area and
    Region Identification}

<Skill Definitions>
- **Direction Adjustment**: Involves turning or changing heading. Look for instructions
    like ''turn left'', ''go back'', or ''face the hallway''. Used when the agent needs to
     rotate or reorient without necessarily changing position.

- **Vertical Movement**: Involves moving across floors or elevation changes. Triggered by
    terms like ''go upstairs'', ''down the stairs'', or ''take the elevator''. Watch for
    floor changes in visuals or references to vertical navigation.

- **Stop and Pause**: Involves coming to a full stop at a defined point. Use lighter-weight
     verbs such as pause, wait, and stand, when the stop happens in the middle of sequence
     (e.g., ''pause by the red sofa''). Use stronger, more terminal verbs like stop and
    come to a stop for the final action or true endpoint (e.g., ''stop at the glass doors
    ''). This distinction helps the agent decide whether to hold briefly or end its
    navigation.

- **Landmark Detection**: Requires identifying and responding to specific objects or
    features in the environment. Triggered by mentions of visible items like ''lamp'', ''
    chair'', ''red sofa'', ''painting''. Used when object recognition is necessary to
    proceed or confirm position.

- **Area and Region Identification**: Involves recognizing or transitioning between
    distinct spaces or rooms. Triggered by mentions like ''enter the kitchen'', ''in the
    bedroom'', ''exit hallway''. Requires understanding of semantic regions based on
    context or appearance.

<Output Format>
Return only the instruction sentence. Do not include tags, labels, or formatting.

---

<Trajectory Images>
''{path_images}''

<Focused Skill>
''{skill_name}''
```

Listing 2: Prompt used for Atomic Skill-specific Data Synthsis

## H.2   PROMPT FOR TEMPORAL REORDERING MODULE

The Temporal Order Module only takes the original natural language instruction as input. It applies the instruction reordering prompt to turn navigation instructions into subgoals $I_{\text{reorder}}$. The prompt is shown in Listing 3, utilizing GPT-4o as the generation model.

```
You are an expert at converting natural language navigation instructions into detailed,
    logically ordered sub-instructions for agents.

<Task>
- Break down instructions into a sequence of minimal, goal-directed steps.
- Make all implicit temporal or spatial relationships explicit.
- Preserve execution order by reconstructing intermediate actions that are implied, not
    directly stated.

<Logic Rules>
- (A) --> [after / then / once / as soon as] --> (B): Do A fully, then B.
- (B) --> [before] --> (A): Move toward A, then perform B at a point prior.
- (A) --> [until] --> (B): Continue A until B is reached.
- Avoid ``then'', ``before'', ``until'', ``once'' etc. in the output.

<Formatting Rules>
- Single sentence, steps separated by periods.
- Each step must be minimal, concrete, and goal-focused.

<Examples>
**Example 1:**
Instruction: ``Turn around and walk down the stairs. Stop once you get down them.''
Output:
Turn around. Walk down the stairs. Stop at the bottom of the stairs.

**Example 2:**
Instruction: ``Walk toward the dining room but turn left before entering it and go into the
    open area.''
Output:
Walk toward the dining room. Stop at the entrance. Turn left. Enter the open area.

**Example 3:**
Instruction: ``After you leave the laundry room, make a left in the hallway, and go to the
    bedroom straight ahead. When you are in the doorway of the room go to the doorway of
    the closet on the left and wait.''
Output:
Exit the laundry room. Turn left in the hallway. Walk to the bedroom straight ahead. Enter
    the doorway of the bedroom. Go to the doorway of the closet on the left. Wait there.

**Example 4:**
Instruction: ``Start moving forward down the corridor. You will pass offices on your left
    and right. Keep going down the hallway until you get to an exit sign on your right and
     what looks like some lockers in front of you. There will also be a brown door with an
     exit sign above it in front of you.''
Output:
Start moving forward down the corridor. Pass the offices on your left and right. Continue
    walking down the hallway. Reach the exit sign on your right and the lockers in front
    of you. Stop in front of the brown door with the exit sign above it.

---

<Original Instruction>:
``{instruction}''
```

Listing 3: Prompt used for Temporal Reordering

### H.3 PROMPTS FOR ACTION ROUTER

The Action Router dynamically selects the most suitable agent at each time step, which can be structured into two distinct reasoning phases: Phase 1 Subgoal Localizer and Phase 2 Skill Router. We provide the detailed prompt for the two phases, respectively. They can be used for either Qwen2.5-VL-7B-Instruct or GLM-4.1V-Thinking-9B .

**Subgoal Localizer.** The Subgoal Localizer identifies the next subgoal to be executed for the current time step and outputs the corresponding reasoning trace. Listing 4 claims the prompt for the subgoal localizer, which takes all reorder subgoals, the previously executed subgoals, and the prior selected viewpoints as input.

**Skill Router.** The skill router determines which skill-based agent is most appropriate for executing the selected subgoal among the 5 skill-based agents. Besides, it receives the original instruction as contextual input to capture additional linguistic cues such as verbs and spatial references. It also uses

```
You are a visual reasoning assistant for indoor navigation.
<Task>:
Your task is to analyze a list of previously observed images and a natural language
    instruction.
Determine which parts of the instruction have already been completed, and return the next
    step to be executed.
<Response Rules>
Your response must:
- Return the next action using *exact phrasing* from the reordered instruction (no
    paraphrasing).
- Match the sub-instruction to the visual context from previous images.
- If the goal (e.g., pool table) has clearly been reached, return the final sub-instruction.

- If *all* sub-instructions have been completed based on the visual path, do not return
    anything further. Stop reasoning.
- If the final destination has been reached and the last step is a positional or waiting
    action (e.g., ``wait there'', ``step to the left''), return that as the next step.
- You must reason about whether the agent is already at the destination.
- If the current image shows the goal destination (e.g., inside the room with the pool
    table, or inside the open doorway), and the instruction contains a final step like ``
    wait'' or ``adjust your position'', that is the next sub-instruction.
---
Use the following reasoning strategy to determine what to do next:
<Step-by-Step Reasoning Instructions>:
1. Decompose the instruction into sub-instructions.
- Break the full instruction into smaller steps. Each sentence or clause typically
    represents one step.
- Example:
    - Original: ``At the bottom of the stairs, go through the nearest archway to your left.
        Head straight until you enter the room with a pool table. Step slightly to the left
        to get out of the way.''
    - Decomposed:
        - ``At the bottom of the stairs, go through the nearest archway to your left.''
        - ``Head straight until you enter the room with a pool table.''
        - ``Step slightly to the left to get out of the way.''
2. Use the previous sub-instruction list to identify completed steps.
- Do not reissue any previously executed sub-instructions.
- Compare upcoming steps against what may have been visually completed, even if not
    explicitly executed one-by-one.
3. Analyze the sequence of previous viewpoint images.
- Use visual context to infer if *multiple* sub-instructions have been completed in a
    single transition.
- If image progression clearly shows the agent has already bypassed an intermediate area or
    reached a later goal, mark those steps as implicitly complete.
4. Evaluate remaining sub-instructions for completion.
- If the current image shows the agent at or beyond the target of a sub-instruction, that
    step can be considered completed.
- If the current image shows the agent inside the goal location and only a final positional
    instruction remains (e.g., ``Step slightly to the left''), return that final
    instruction.
5. Select the next uncompleted sub-instruction that is visually and contextually justified.
- Use exact wording from the original instruction.
- Do not return instructions that the agent already visually fulfilled, even if they were
    skipped.
6. Output the result in the following JSON format:
{
"Sub-instruction to be executed": "<exact next instruction clause>",
"Reasoning": "<why this is the next step based on image sequence>"
}
CHECKPOINT:
If multiple sub-instructions were completed based on a single or continuous image segment,
    skip them and jump to the next logical, visually unfulfilled step.
---

Now, using the instruction and the visual history, identify the next step.
IMPORTANT: Your response must be a valid JSON object without any surrounding text, code
    blocks, or explanations.
Do not include markdown formatting like ```json or ```.

<Original Whole Instruction>:
``{instruction}''
<Previous Sub-Instructions>:
``{previous_sub_instructions}''
<Previous Viewpoint Images>:
```

Listing 4: Prompt used for Subgoal Localizer in Action Router

the reasoning trace from the subgoal localizer to enhance its understanding of the current subgoal. The whole process is displayed in Listing 5.

```
You are a visual reasoning assistant for indoor navigation.

<Available Skills>:
[''Direction Adjustment'', ''Vertical Movement'', ''Stop and Pause'', ''Landmark Detection
    '', ''Area and Region Identification'']

<Skills Explanation>:
- Direction Adjustment:
Involves turning or changing heading. Look for instructions like ''turn left'', ''go back
    '', or ''face the hallway''. Used when the agent needs to rotate or reorient without
    necessarily changing position.
- Vertical Movement:
Involves moving across floors or elevation changes. Triggered by terms like ''go upstairs
    '', ''down the stairs'', or ''take the elevator''. Watch for floor changes in visuals
    or references to vertical navigation.
- Stop and Pause:
Involves stopping at a specific location. Triggered by instructions like ''stop'', ''wait
    '', or ''stand in front of''. Used when the endpoint or a mid-action pause is
    important.
- Landmark Detection:
Requires identifying and responding to specific objects or features in the environment.
    Triggered by mentions of visible items like ''lamp'', ''chair'', ''red sofa'', ''
    painting''. Used when object recognition is necessary to proceed or confirm position.
- Area and Region Identification:
Involves recognizing or transitioning between distinct spaces or rooms. Triggered by
    mentions like ''enter the kitchen'', ''in the bedroom'', ''exit hallway''. Requires
    understanding of semantic regions based on context or appearance.

<Task>:
1. Read and understand the sub-instruction to be executed.
2. Use the reasoning explanation to infer what skills are likely required to carry out that
     sub-instruction.
3. Choose the top 1 skill that is most relevant to the sub-instruction.

<Input>:
You will be given:
- The original full navigation instruction.
- The sub-instruction that should be executed next, based on reasoning.
- A reasoning explanation derived from the visual history and instruction.

Output exactly **one skill name** from the above list.
Do not provide explanations or additional text.

<Output Format>:
*****SKILL_NAME*****

<Example>
Original Whole Instruction: ''At the bottom of the stairs, go through the nearest archway
    to your left. Head straight until you enter the room with a pool table. Step slightly
    to the left to get out of the way.''

Sub-instruction to be executed for next step: ''Head straight until you enter the room with
     a pool table.''

Reasoning based on previous viewpoints path and original instruction: The agent appears to
    be just outside the archway. The next step is likely to involve entering the archway
    and preparing to head straight.

Expected Output:
*****Landmark Detection*****

---

<Reordered Whole Instruction>:
''{full_instruction}''

Sub-instruction to be executed for next step:
''{sub_instruction}''

<Reasoning based on previous viewpoints path and original instruction>:
''{reasoning}''
```

Listing 5: Prompt used for Skill Router in Action Router

