# OpenReview forum: "Breaking Down and Building Up: Mixture of Skill-Based Vision-and-Language Navigation Agents"
_ICLR.cc/2026/Conference — ICLR 2026 Conference Withdrawn Submission_

### Official Review · Reviewer_WmQk · 2025-10-29

**Soundness:** 3
**Presentation:** 4
**Contribution:** 2
**Rating:** 4
**Confidence:** 5

**Summary:**

This paper proposes SkillNav, a mixture-of-experts framework designed to decompose vision-and-language navigation (VLN) into atomic skills and route-level reasoning. The approach aims to improve compositional generalization by assigning navigation instructions to specialized experts and reordering sub-instructions using LLMs. The authors also introduce several skill-based datasets to support the design and use of VLMs as the router for different experts. Experiments on R2R and GSA-R2R demonstrate the effectiveness of the proposed SkillNav especially in GSA-R2R with styled instructions.

**Strengths:**

1. Interesting motivation and idea. The paper presents a clear and appealing idea of decomposing navigation into modular “skills” and routing instructions through specialized experts. This formulation aligns well with the broader goal of enhancing compositional reasoning in VLN. The authors also provide distinct datasets for training each expert, which adds practical value and can facilitate future research in this direction.
2. The proposed SkillNav framework effectively combines the generalization ability of LLM-based methods with the strong task-specific performance of supervised VLN models through a hierarchical structure. This hybrid design is promising and represents a good balance between accuracy and efficiency.
3. Clear and well-organized presentation. The paper is well-written and easy to follow. The motivation, methodology, and experiments are presented in a coherent and logical flow, making the main contributions easy to understand and evaluate.

**Weaknesses:**

1. Misalignment between motivation and method. The motivation and the proposed method don’t quite line up. While the paper claims to tackle compositional reasoning through an MoE setup, the approach feels more like a form of data augmentation. The results in Table 4 also suggest that the MoE component doesn’t really make a difference — the no-router variant performs almost the same. The largest improvement happens in the Test-N-Scene split of GSA-R2R, but the method doesn’t include any mechanism specifically designed for handling scene-style instructions. This improvement might actually come from the reordering module, where the LLM helps better interpret the instructions, rather than from the MoE design itself.
2. Modest experimental gains and potential data leakage. The experimental results are not very convincing. On the standard R2R benchmark, the method falls short compared to SRDF, which also augments instructions in a similar way. The claimed improvement mainly comes from GSA-R2R, but that dataset includes scenes from HM3D, which are also used in training by ScaleVLN and the proposed SkillNav. This overlap raises a concern about possible data leakage, which could partly explain the performance gains.
3. Missing ablations and unclear MoE behavior. The paper would benefit from more ablation studies to verify whether the MoE setup actually works as intended. Right now, the model uses VLMs to directly predict the skill without any fine-tuning, which is odd given that the paper also introduces a skill-labeled dataset. It’s unclear why that dataset wasn’t used to train or adapt the experts. In addition, there’s no analysis showing how the routing mechanism selects the right expert for a given instruction. Without this, it’s hard to tell whether the model is really leveraging multiple experts or just behaving like a black box.

**Questions:**

1. In Lines 48–52, the authors state that existing methods tend to memorize examples, limiting their effectiveness in unseen environments. Could the authors provide evidence for this claim? Some recent works, such as ScaleVLN and SRDF, have already narrowed the performance gap between val-seen and val-unseen, which seems to contradict that statement.
2. More explanation is needed regarding the “atomic skills” derived from NavNuances. Since these form the foundation of the proposed method, readers would benefit from a clearer understanding of what each skill represents and how they are defined or annotated.
3. In Table 3, the landmark detection performance is noticeably lower than the other subtasks. Could the authors elaborate on why this happens? Does it relate to data imbalance, annotation difficulty, or inherent limitations of the current detection pipeline?

---

> ### Author Response · Authors · 2025-12-04
> **Dealing with the concern about Misalignment between motivation and method**
>
> We sincerely appreciate the reviewer’s acknowledgment of our strengths and contributions. We address the reviewer’s comments as follows:
>
> **1. SkillNav is Not Data Augmentation**
>
> We clarify that SkillNav is not a data augmentation method, even though it uses synthetic skill-specific data for training individual skill agents. The role of synthetic data in our framework is to enable explicit specialization of atomic skills, not to improve a single monolithic policy. The core contribution of SkillNav is the explicit decomposition of navigation into atomic skills and the dynamic recomposition of these skills at inference time through a structured MoE routing mechanism. This is fundamentally different from data augmentation, which still relies on a single end-to-end policy without modular execution or dynamic expert selection.
>
> To empirically address this concern, we conducted a controlled experiment (Table A and Table B) comparing SkillNav against a **"Mixed Skills"** baseline: a single monolithic DUET agent fine-tuned on the union of all synthetic skill data used in SkillNav.
>
> If the performance gains were driven solely by data augmentation, the "Mixed Skills" agent should match SkillNav’s performance. However, our results show the opposite, confirming that structural decomposition (MoE), not data volume, is the driver of improvement.
>
> As shown in **Table A**, the "Mixed Skills" agent achieves an SR of $54.26\%$ on **Test-N-Scene**, which is statistically indistinguishable from the baseline ScaleVLN ($54.50\%$). In stark contrast, SkillNav achieves **$56.66\%$ SR** ($+2.4\%$ absolute gain). This gap confirms that simply exposing a model to compositional data is insufficient. The model requires the MoE architecture to dynamically route sub-problems to specialized experts rather than forcing a single network to learn conflicting behaviors.
>
> **Table A: GSA-R2R DUET agent performance (SR/SPL) on Test-R-Basic, Test-N-Basic, and Test-N-Scene.** *Note: Data augmentation alone (Mixed Skills) fails to improve over the baseline.*
>
> | **Agent**              | **Test-R-Basic (SR / SPL)** | **Test-N-Basic (SR / SPL)** | **Test-N-Scene (SR / SPL)** |
> | ---------------------- | --------------------------- | --------------------------- | --------------------------- |
> | ScaleVLN               | 78.12 / 66.70               | 69.13 / 57.16               | 54.50 / 42.70               |
> | Mixed Skills           | 77.60 / 66.76               | 70.23 / 59.32               | 54.26 / 43.68               |
> | Direction Adjustment   | 78.80 / 68.40               | 70.88 / 59.53               | 53.48 / 42.22               |
> | Vertical Movement      | 78.42 / 67.52               | 70.58 / 59.22               | 54.57 / 43.78               |
> | Landmark Detection     | 78.56 / 67.84               | 70.41 / 59.65               | 54.27 / 44.77               |
> | Area and Region Ident. | 78.22 / 68.24               | 71.27 / 60.81               | 54.60 / 44.93               |
> | Stop and Pause         | 78.38 / 67.93               | 70.03 / 58.77               | 53.44 / 42.02               |
> | **SkillNav**           | **78.83 / 68.88**           | **71.58 / 61.34**           | **56.66 / 47.96**           |
>
> **Table B** illustrates why the monolithic approach fails and why our MoE method aligns with the motivation of compositional reasoning and shows the necessity of specialization.
>
> This demonstrates that SkillNav functions as a true MoE system: it leverages agents that have "overfit" to specific atomic skills (high performance in their domain, lower in others) and uses the router to compose them. A data augmentation approach cannot replicate this sharpness in skill execution.
>
> **Table B: Evaluation of skill-based agents on the NavNuances benchmark.** *Categories: Direction Change (DC), Vertical Movement (VM), Landmark Recognition (LR), and Room Recognition (RR). VM includes SR/OSR/SPL; RR reports SR/OSR.*
>
> | **Methods**          | **DC (SR)** | **VM (SR / OSR / SPL)**       | **LR (SR)** | **RR (SR / OSR)** |
> | -------------------- | ----------- | ----------------------------- | ----------- | ----------------- |
> | Mixed Skills         | 66.84       | 84.11 / 87.65 / 79.22         | **48.90**   | 81.82 / 90.91     |
> | Direction Adjustment | **70.81**   | 81.76 / **91.18** / 76.28     | 31.39       | 81.82 / 94.91     |
> | Vertical Movement    | 70.68       | **87.65** / 89.41 / **83.83** | 30.22       | 82.18 / 96.00     |
> | Landmark Detection   | 70.29       | 82.35 / 85.29 / 78.94         | 31.53       | 83.64 / **97.09** |
> | Area/Region Ident.   | 67.53       | 84.12 / 88.82 / 80.49         | 29.20       | **85.09** / 96.36 |
> | Stop and Pause       | 68.91       | 84.71 / 87.06 / 80.67         | 29.78       | 83.64 / **97.09** |

---

> > ### Author Response · Authors · 2025-12-04
> > **Novel Instruction Handling Mechanism in SkillNav**
> >
> > We thank the reviewer for this insightful observation. We clarify that the improvement on **Test-N-Scene** does not come from the Temporal Reordering module alone, nor is it due to any scene-specific instruction handling built into the model. Instead, the gain arises from the joint effect of structured temporal planning and Mixture-of-Skills execution, which together enable SkillNav to better handle complex, long-horizon, and stylistically diverse instructions.
> >
> > The **Temporal Reordering Module** plays a structural role inside the routing pipeline. It converts free-form instructions into an explicitly ordered subgoal sequence and supplies this sequence **only to the action router for subgoal localization**. Importantly, it does not execute actions, nor does it provide any extra semantic supervision to the skill agents. All skill-based agents still operate solely on the original full instruction and visual observation. Therefore, temporal reordering alone cannot account for the navigation improvement.
> >
> > The **Action Router** is responsible for mapping each localized subgoal to the appropriate specialized skill agent. Without the MoE routing, the system cannot translate the reordered plan into skill-conditioned execution. This is directly supported by our ablation results: when routing is random or misaligned, performance drops even with temporal reordering enabled. Conversely, when both reordering and structured MoE routing are enabled, performance becomes stable and significantly improves, especially on Test-N-Scene.
> >
> > In summary, the performance gains on Test-N-Scene cannot be attributed to the reordering module alone. Temporal reordering reduces ambiguity in planning, while the MoE router is essential for translating each subgoal into the correct expert action. The improvement comes from their tight integration, validating that the gain is fundamentally due to structured compositional reasoning via skill recomposition, rather than instruction parsing alone.
> >
> > Here is a qualitative example showing routing on a scene-style instruction:
> >
> > consider a typical narrative instruction from GSA-R2R Test-N-Scene:
> >
> > > “Alright folks, let’s take a gentle stroll down these beautiful marble steps here. Keep going straight ahead, and you’ll notice those magnificent stone pillars on your left. Just keep moving forward until you see the wooden confessionals and, right beyond them, those impressive large double doors. That’s where we’ll pause.”
> >
> > The Temporal Reordering module first distills this narrative into discrete subgoals (e.g., *“Begin walking down the marble steps,” “Observe the stone pillars,”* etc.). However, SkillNav does not rigidly execute this list; instead, it dynamically routes control based on the visual environment.
> >
> > Execution begins when the router detects the stair structure and assigns the current subgoal to the **Vertical Movement** skill. Crucially, this agent receives the **original full instruction**. Upon activation, it parses the complete narrative to locate and attend to the specific phrase *“down these beautiful marble steps,”* grounding this text to the visual descent while ignoring later landmark descriptions.
> >
> > Once the agent descends and aligns with the corridor, the router shifts control to the **Direction Adjustment** skill. This agent processes the full instruction to isolate the directive *“keep going straight ahead,”* executing the forward movement. Subsequently, when the router activates the **Landmark Detection** skill for the "stone pillars" subgoal, the agent again references the original text. It attends to the detailed description—*“magnificent stone pillars on your left”*—to perform precise visual grounding, ensuring the agent identifies the specific object mentioned in the tour.
> >
> > Finally, the router tracks the destination using the **Stop and Pause** skill. This agent verifies the visual presence of the doors against the instruction's final clause (*“that’s where we’ll pause”*) before terminating the episode. Throughout the trajectory, the router dictates *which* expert is active, but the experts themselves derive their visual-linguistic understanding directly from the rich, original instruction.

---

> ### Author Response · Authors · 2025-12-04
> **Response to Data Leakage Concern**
>
> The reviewer raises a valid concern that the improvements on GSA-R2R may be influenced by potential data leakage due to shared HM3D scenes and instruction augmentation. To address this concern rigorously, we analyze possible leakage from two complementary perspectives: (1) *vision-level scene overlap* through HM3D-based augmentation, and (2) *language-level instruction overlap* between training data and the GSA-R2R test sets. We show that the HM3D overlap is shared by all competing methods and does not uniquely benefit SkillNav, and that there is no meaningful language-level test leakage from either ScaleVLN or our synthetic datasets.
>
> **Vision-Level Scene Data Overlap with HM3D.**
>
> It is correct that each skill agent in SkillNav is fine-tuned using ScaleVLN augmentation data, which includes 899 HM3D scans. GSA-R2R also contains scenes from HM3D.
>
> However, this situation is not unique to SkillNav. Both SRDF and ScaleVLN use similar HM3D-based visual augmentation. Therefore, any potential benefit from HM3D overlap is shared by all these methods and does not give SkillNav an unfair advantage over SRDF. Moreover, our main contribution is not the use of HM3D itself, but the skill-level decomposition and compositional training strategy.
>
> Importantly, our skill synthetic trajectories are all constructed from the seen split of MP3D R2R, not from GSA-R2R. This avoids direct trajectory or instruction leakage from the GSA-R2R test set.
>
> **No Language-Level and Instruction Leakage.**
>
> We first analyze the vocabulary overlap between the **ScaleVLN augmentation instructions** and the **GSA-R2R test instructions**, since ScaleVLN is used for vision-level fine-tuning and is the primary potential source of language leakage.
>
> All instructions are lowercased and tokenized using the regular expression (`[a-zA-Z']+`) to construct unique vocabularies. Under this protocol, the ScaleVLN augmentation file contains **208 unique tokens**. Although ScaleVLN includes 2,891,134 instruction entries, these instructions are generated by a constrained template-based pipeline with a fixed pool of action verbs, spatial prepositions, and object categories. As a result, the number of unique token types remains small despite the large number of instruction instances.
>
> ScaleVLN vs. GSA-R2R Vocabulary Overlap.
>
> The table below reports the vocabulary overlap between ScaleVLN augmentation instructions and the three GSA-R2R test splits. While ScaleVLN covers a large fraction of its own limited vocabulary in GSA-R2R, the coverage of the GSA-R2R test vocabulary by ScaleVLN remains low. In particular, across the union of all three GSA-R2R test sets, only 196 out of 1,645 tokens overlap, corresponding to 11.9% test vocabulary coverage. This indicates that the majority of GSA-R2R test words are not observed in ScaleVLN augmentation.
>
> **Table 3: Vocabulary overlap between ScaleVLN augmentation instructions and GSA-R2R test sets.**
>
> | **GSA-R2R Test Split** | **Overlap** | **Test Vocab Size** | **Coverage (%)** |
> | ---------------------- | ----------- | ------------------- | ---------------- |
> | Residential Basic      | 186         | 514                 | 36.2             |
> | Non-Residential Basic  | 166         | 578                 | 28.7             |
> | Non-Residential Scene  | 168         | 1,376               | 12.2             |
> | **Union of All Tests** | **196**     | **1,645**           | **11.9**         |
>
> We further report the vocabulary coverage of our GPT-4o generated synthetic skill datasets with respect to the union of all GSA-R2R test vocabularies. The atomic skill coverage ranges from **21.95% to 27.72%** across different skill types. Even in the highest case (Temporal Order Planning Skill), more than 58% of the GSA-R2R test vocabulary remains unseen. This confirms that our synthetic language does not duplicate the GSA-R2R test distribution.
>
> Taken together, these results demonstrate that the performance gains on GSA-R2R cannot be explained by either vision-level or language-level data leakage from HM3D, ScaleVLN, or our synthetic annotations. Instead, the improvements stem from the proposed skill decomposition and compositional learning framework.

---

> ### Author Response · Authors · 2025-12-04
> **Explanation on Atomic Skill Definitions and Annotations and Lowest Success Rate of Landmark Recognition in NavNuance**
>
> **Atomic Skill Definitions and Annotations**
>
> We adopt four atomic skills from the NavNuances framework that are essential for physical navigation and spatial grounding. Each skill is rigorously defined using geometric and semantic constraints to ensure it represents a minimal, isolatable capability.
>
> It is crucial to distinguish between the **semantic intent** and the **motor execution**: **Semantic Level (Atomic):** A skill corresponds to a single, indivisible navigation intent (e.g., "turn around," "go upstairs," "find the sofa"). **Execution Level (Multi-step):** A single intent may require multiple discrete actions in the simulator.
>
> For example, consider the instruction **“Walk to the far end of the room.”** Semantically, this is a single **Area and Region Identification** task. Physically, the agent may need to adjust its heading, take three forward steps, and stop. Because all these low-level actions serve one unified semantic goal, they constitute a single atomic skill. An instruction only becomes "multi-skill" if it introduces a second independent goal (e.g., "...and then turn right").
>
> | **Skill Category**               | **Definition & Examples**                                    | **Key Constraints & Criteria**                               |
> | -------------------------------- | ------------------------------------------------------------ | ------------------------------------------------------------ |
> | **Direction Adjustment**         | Captures pure changes in heading.  **Ex:** *"turn left," "face the wall"* | • **Metric:** Defined by the angle difference between viewpoints. • **Magnitude:** Requires a significant shift, but **excludes U-turns** to avoid ambiguity. • **Normalization:** Vague terms (e.g., *"slightly turn"*) are mapped to their actual geometric change. |
> | **Vertical Movement**            | Models motion between floor levels.  **Ex:** *"go upstairs"* | • **Threshold:** Absolute height difference must be **$> 2$ units** (filters out slopes). • **Structure:** Must involve stairs or elevators. • **Consistency:** Movement must be strictly unidirectional (only up or only down). |
> | **Landmark Detection**           | Focuses on visually grounded object navigation.  **Ex:** *"walk towards the sofa"* | • **Visibility:** Target must be a visually distinct instance (not just a category). • **Persistence:** The object must remain visible across **multiple steps**, ensuring the agent is tracking it continuously. |
> | **Area & Region Identification** | Models semantic transitions between functional areas.  **Ex:** *"go into the bedroom"* | • **Transition:** Requires an explicit change in the semantic region label. • **Plane:** Restricted to **horizontal movement** only (no vertical changes). • **Criteria:** Success is based on being **inside** the room (inclusion), not just distance to a point. |
>
> Note: We exclude Numerical Comprehension (e.g., "count the third door") as it tests memory rather than motion control. This is handled implicitly by our higher-level router.
>
> These skills are "atomic" because they isolate exactly one capability—heading, floor change, object grounding, or room transition—without mixing concepts. In a discrete navigation graph, they are operationally independent.
> For example, "Go through the double doors" activates only **Landmark Detection** because the agent selects the viewpoint containing the doors; it does not require a separate "Direction Adjustment" skill to steer.
>
> **Explanation on Lowest Success Rate of Landmark Recognition in NavNuances**
>
> The lower performance in the **Landmark Recognition (LR)** category in NavNuances stems from the inherent difficulty of fine-grained instance grounding rather than simple object detection. We identify three primary challenges:
>
> 1. **Spatial Reasoning Complexity:** Models struggle with relational concepts like **“walk past.”** Failure cases show agents often stop *beside* an object, mistaking proximity for completion, or confuse the front/back views. This suggests the bottleneck is not recognizing the object, but understanding the temporal geometric condition of "passing" it.
> 2. **Instance-Level Precision:** The task requires distinguishing a specific instance (e.g., *this* specific chair) rather than just a category.
> 3. **Viewpoint Sparsity:** In discrete graphs (Matterport3D), landmarks are often visible from very few viewpoints. A minor error in viewpoint selection can result in a large geometric penalty in the distance-based evaluation, turning small perception errors into task failures.
>
> The difficulty lies in reasoning, not just vision. Stronger VLMs improve on "walk towards" (recognition dominant) but still struggle with "walk past" (reasoning dominant). This motivates our SkillNav design: using a VLM as a high-level reasoner to handle these complex spatial relationships, rather than relying solely on a low-level policy.

---

> ### Author Response · Authors · 2025-12-04
> **Evidence of Supervised Agents' Memorization**
>
> While recent methods like ScaleVLN and SRDF have indeed narrowed the performance gap on R2R splits (val-seen vs. val-unseen), this improvement does not reflect true generalization. Instead, our analysis suggests these methods rely on linguistic memorization (due to high distribution overlap) and task-specific overfitting (via augmentation construction), causing them to fail in rigorous generalization benchmarks like GSA-R2R.
>
> **Linguistic Memorization due to Distributional Homogeneity**
>
> The "unseen" environments in R2R benchmarks are not linguistically novel. Our analysis of the R2R instruction vocabulary reveals that the distribution of instructions in validation splits is statistically indistinguishable from the training distribution.
>
> - There is a Jaccard **similarity score of 0.9992** between instruction sets, with effectively perfect token coverage.
> - The Cosine similarity of token frequency vectors is 1.0, and the Jensen–Shannon divergence is negligible ($8.2 \times 10^{-6}$).
>
> This implies that the "narrowing gap" the reviewer observes is likely due to models memorizing these static linguistic patterns, which remain constant across splits, rather than learning to ground instructions in truly unseen visual environments.
>
> **Failure in Robust Generalization Benchmarks (GSA-R2R)**
>
> To test if the performance gap reduction translates to real-world robustness, we evaluated these methods on GSA-R2R, a benchmark designed to test generalization beyond R2R splits.
>
> - Despite high scores on common-used R2R, both ScaleVLN and SRDF exhibit significant performance drops in GSA-R2R testing.
> - Crucially, this failure persists even when the augmentation data includes scenes that were technically "seen" during training. This indicates that their success on R2R is brittle and relies on specific dataset biases that do not hold in broader generalization scenarios.
>
>
>
> **Methodological Memorization via Data Construction**
>
> Finally, we argue that the definition of "memorization" extends to the training methodology.
>
> - Methods like SRDF require constructing specific augmentation data tailored for each new task.
> - This reliance on constructing task-specific priors acts as a form of structural memorization. The model is not generalizing to a new task zero-shot; it is overfitting to the constraints provided by the engineered augmentation data.

---

> ### Author Response · Authors · 2025-12-04
> **Human Evaluation on Skill Routing**
>
> To address the concern regarding the routing mechanism's decision-making process, we conducted a detailed human evaluation of failure cases in the R2R Val-Unseen split. By analyzing the specific steps where the agent failed, we provide evidence that the router is not behaving as a "black box," but rather as a linguistically competent planner that is primarily bottlenecked by visual grounding limitations.
>
>
> **Human Evaluation of Failure Routing**
>
> We examined 17 distinct failure cases where specific step-level data was available.
> Quantitatively, the average success rate of the routing mechanism, defined as the proportion of steps completed successfully before the first failure, is 24.5%.
> This indicates that while the agent effectively initiates navigation, critical routing errors tend to occur in the early-to-mid stages of the trajectory.
>
>
> **Skill Selection and Visual Grounding**
>
> The core analysis reveals that the router's failures are not random but instead exhibit a clear pattern of visual grounding deficits. As shown in the table below, there is a significant discrepancy between the skills selected by the router and those actually required by the environment.
>
> The *Landmark Detection* expert is severely under-utilized, being the optimal skill in **76% (13 of 17)** of the analyzed failures, yet the router correctly selected it in only one instance. Instead of activating the *Landmark* expert to identify specific objects—such as a curio cabinet or bed—the router frequently defaulted to coarser navigational skills like *Region Identification* (6 cases) or *Directional Adjustment* (5 cases).
>
> This mismatch indicates that while the router successfully interprets the linguistic instruction to move or change areas, it fails to bind the instruction to the specific visual target required to trigger the *Landmark* expert. This confirms that the VLM experts still struggle to bind visual object references effectively even with Temporal Reordering in place, suggesting that the primary error source is visual grounding rather than high-level planning logic.
>
>
> **Table: Confusion matrix between Router Selection vs. Human Ground Truth**
>
> *Note: Numbers denote counts; percentages are row-normalized. The high concentration of failures in the **Landmarks** column reveals a visual-grounding gap.*
>
> | **Selected Skill** | **Direction (Required)** | **Landmarks (Required)** | **Region (Required)** |
> | ------------------ | ------------------------ | ------------------------ | --------------------- |
> | **Direction**      | 2 (40%)                  | 2 (40%)                  | 1 (20%)               |
> | **Landmarks**      | 0                        | 1 (100%)                 | 0                     |
> | **None**           | 0                        | 2 (100%)                 | 0                     |
> | **Region**         | 0                        | 6 (100%)                 | 0                     |
> | **Stop**           | 0                        | 2 (67%)                  | 1 (33%)               |
>
> **Failure Modes Analysis**
>
> Qualitative analysis identifies several specific mechanisms by which these failures manifest:
>
> - **Initial Viewpoint Misalignment:** A frequent cause of early failure is where the agent starts with an orientation that does not align with the instruction; this initial drift often propagates downstream as the router struggles to recover from an incorrect starting heading.
> - **Hallucinations:** These play a significant role in derailing routing. For example, the agent generated a subgoal to *"Position yourself beside the glass case with the dolls"* when the instruction simply stated *"Enter the bedroom,"* illustrating that the action router can create irrelevant descriptions grounded in neither the text nor the view, which subsequently propagates incorrect skill choices.
> - **Controller Limitations:** Even when the router selects the correct skill, execution can fail. For instance, valid *Direction* transitions failed when multiple direction subgoals were stacked (e.g., *"Turn left. Turn right. Turn left again."*), indicating that the skill-conditioned controllers themselves can become a bottleneck during complex sequences.
> - **Termination Errors:** Finally, the router occasionally fails to generate necessary "Stop" subgoals or struggles with ambiguous spatial prepositions, leading to incorrect termination points.
>
> In conclusion, the analysis demonstrates that the routing mechanism is not acting as a random black box. It exhibits strong instruction understanding capabilities—often sequencing subgoals correctly in text—but acts as a bottleneck when it relies on textual priors over visual evidence. The frequent substitution of *Region/Direction* skills for *Landmark* skills highlights that improving the router's ability to attend to visual object cues is the most critical path for future performance gains.

---

### Official Review · Reviewer_EJ2y · 2025-10-31

**Soundness:** 3
**Presentation:** 3
**Contribution:** 2
**Rating:** 4
**Confidence:** 3

**Summary:**

SkillNav is a modular VLN framework that decomposes navigation into atomic skills, uses an LLM to reorder instructions into subgoals, and employs a VLM-based router to pick the right skill at each step. Each skill has a synthetic, skill-focused dataset and a specialized agent fine-tuned on a DUET backbone, then all agents are integrated for execution. The method attains strong R2R results and state-of-the-art generalization on GSA-R2R, and shows skill-wise gains on NavNuances.

**Strengths:**

1. Clear modular design with interpretable skills, temporal reordering, and a VLM router that localizes subgoals and selects a single best skill.
2. Practical synthetic data pipeline that enables skill-specific supervision without human annotation.
3. Reasonable empirical results on GSA-R2R with competitive R2R performance and skill-level improvements on NavNuances.

**Weaknesses:**

1. It would be great to see if the method can generalize to a broader setting, such as real-world robotic settings or computer-use agent settings. The current evaluations on VLN tasks are somewhat limited and artificial.
2. Compared with other baselines such as SRDF, the model still falls behind and there is a significant performance gap between their model and other baselines on benchmarks.
3. Router effectiveness depends on the chosen VLM, and ablations show nontrivial variance across routers.

**Questions:**

1. Can the model fit into the current MLLM-based paradigm?

---

> ### Author Response · Authors · 2025-12-04
> **Generalization to broader settings and Performance Gap with SRDF**
>
> We appreciate the reviewer's recognition of our clear modular design and contributions. We address the reviewer’s comments as follows:
>
> **Generalization to broader settings**
>
> We agree that evaluating generalization beyond simulator-based VLN benchmarks is important. In this work, we evaluate our method on widely used VLN benchmarks that are common-used for both supervised agents and LLM-based methods. These benchmarks enable controlled and fair comparison with prior work, though they indeed remain limited in reflecting real-world deployment settings. More recent datasets such as **GSA-R2R** introduce more diverse scenes and novel instruction styles, and our strong performance on this benchmark already provides evidence for improved generalization beyond VLN settings.
>
> We note that **DiscussNav** includes a real-world demonstration setting, and extending our framework to such real-world embodied platforms is an important direction for future work. We also discuss other continuous-environment VLN settings such as **VLN-CE**, which bridge perception and control but still share similar structures and instruction styles with simulated benchmarks.
>
> Beyond embodied navigation, our skill decomposition framework is closely related to recent advances in computer-use and GUI agents. For example, **Mirage-1**[1] introduces a *Hierarchical Multimodal Skills (HMS)* module that explicitly abstracts trajectories into multiple levels, including execution skills, core skills, and meta-skills. This represents an even more explicit form of skill-level abstraction. Our method follows a similar principle by decomposing complex behaviors into reusable atomic skills and dynamically coordinating them through a high-level router. This close alignment with hierarchical skill-based GUI agents further supports that our approach captures general, transferable principles of multimodal decision-making and is naturally extensible across different embodied and computer-use agent settings.
>
> [1] Yuquan Xie, Zaijing Li, Rui Shao, Gongwei Chen, Kaiwen Zhou, Yinchuan Li, Dongmei Jiang, and Liqiang Nie. Mirage-1: Augmenting and updating gui agent with hierarchical multimodal skills, 2025. URL https://arxiv.org/abs/2506.10387.
>
> **Explanation on Performance Gap with SRDF**
>
> While **SRDF** achieves strong performance on the in-distribution R2R benchmark, it shows clear limitations in cross-domain generalization. As shown in our results, although SRDF slightly outperforms SkillNav on R2R *Val-Unseen* and *Test-Unseen*, its performance drops noticeably when evaluated on the more challenging **GSA-R2R** benchmarks, which test generalization to new environments and instruction styles.
>
> On the commonly used R2R benchmark itself, SkillNav surpasses all other baselines except SRDF across all evaluation metrics, establishing it as one of the strongest methods even in the in-distribution setting. More importantly, SkillNav is the **best generalizable method** across diverse environments and instruction styles, achieving strong cross-dataset performance without any additional retraining. This directly validates the effectiveness and extensibility of our skill-based design.
>
> In contrast, SkillNav consistently outperforms all baselines, including SRDF, across every GSA-R2R split. Specifically, SkillNav achieves:
>
> - **+13% higher Success Rate (SR)** on *Test-N-Basic*.
> - **+5% higher Success Rate (SR)** on *Test-N-Scene*.
> - The **best SPL** across all GSA-R2R splits.
>
> These results demonstrate that the performance gap between SkillNav and SRDF is limited to the in-distribution R2R setting and does not reflect true generalization ability. SkillNav demonstrates both higher success and more efficient navigation when facing diverse distribution shifts.

---

> ### Author Response · Authors · 2025-12-04
> **Explanation of Effectiveness of the Router**
>
> We emphasize that the router is an essential design component in SkillNav, but an action router alone is not sufficient. The main performance gain of SkillNav comes from the combination of explicit skill decomposition, specialized skill-based agents, and structured routing.
>
> As shown in **Table 4** (Ablation results on GSA-R2R in the main paper), using a learned VLM-based action router consistently outperforms random routing across all GSA-R2R splits. This directly verifies that accurate skill selection is critical for effective mixture-of-skills navigation. At the same time, these results also show that routing must be properly structured to be reliable.
>
> We further observe that different zero-shot VLMs share similar limitations when they are directly used as skill routers. The variance across routers mainly reflects their differences in visual grounding and reasoning ability, but without additional structure, none of them is sufficiently strong to robustly perform both subgoal localization and skill selection from raw instructions.
>
> This is clearly evidenced by the consistently worse performance of all routers when the **Temporal Reordering** module is disabled, as reported in **Table X** below. Although different zero-shot VLMs are used as routers, they all exhibit similar performance ceilings when operating directly on the raw instruction without structured subgoals. This confirms that an action router alone is not sufficient, and that structured temporal guidance is necessary for stable and accurate routing.
>
> **The Role of Temporal Reordering** The Temporal Reordering Module acts as a lightweight teacher. It reorganizes the original instruction into a temporally ordered subgoal sequence and provides this structured plan as input to the action router in Phase 1. This effectively reduces the complexity of the routing problem, significantly stabilizing skill selection. Importantly, the output of the reordering module is **only used by the action router**. All skill-based agents still receive and operate solely on the original full instruction, ensuring that their decision-making remains independent of this auxiliary supervision.
>
> With the Temporal Reordering module enabled, the performance gap across different routers becomes small, demonstrating that once proper temporal structure is provided, SkillNav becomes robust to the specific choice of VLM router, and its overall performance is primarily governed by the skill-based agents.
>
> **Table X: SkillNav's Router performance comparison on GSA-R2R Test-R-Basic, Test-N-Basic and Test-N-Scene when the Temporal Reordering module is disabled.**
>
> | **Action Router**      | **Test-R-Basic (SR / SPL)** | **Test-N-Basic (SR / SPL)** | **Test-N-Scene (SR / SPL)** |
> | ---------------------- | --------------------------- | --------------------------- | --------------------------- |
> | Qwen2.5-VL-7B-Instruct | 78.42 / 67.80               | 71.01 / 59.62               | 55.46 / 45.43               |
> | GLM-4.1V-9B-Thinking   | 77.46 / 66.27               | 70.70 / 58.63               | 55.62 / 42.64               |
> | InternVL3.5-8B         | 78.69 / 67.40               | 72.07 / 60.49               | 55.65 / 46.94               |

---

> ### Author Response · Authors · 2025-12-04
> **How to Fit into MLLM-based Paradigm**
>
> **SkillNav is already compatible with the MLLM-based paradigm.**
>
> SkillNav naturally fits into the MLLM-based framework because the router itself is a VLM or MLLM that performs high-level multimodal reasoning and decides which skill to activate at each step. In this sense, the MLLM plays the role of a **high-level controller**, while the skill-based agents serve as modular **low-level policy executors**. This design follows the common MLLM paradigm of using large foundation models for reasoning and lighter modules for control, but in a more structured and interpretable way through explicit skill decomposition.
>
> **Why the skill-based agents are not VLM-based models.**
>
> The SkillNav framework itself is model-agnostic and only assumes the availability of specialized skill executors. In principle, these skill executors could be instantiated by any policy model. In our implementation, however, we deliberately use pretrained supervised VLN backbones rather than VLMs for skill execution, based on both effectiveness and efficiency considerations.
>
> **Effectiveness:** Existing LLM or VLM-based VLN agents such as MapGPT, NavGPT-2, VLN-R1, and DiscussNav consistently lag behind supervised VLN models on R2R. More importantly, they exhibit unstable generalization on GSA-R2R. This shows that directly using MLLMs as low-level action policies is still unreliable for precise and robust navigation, especially in unseen environments.
>
> **Efficiency:** Our analysis in Section 5.3 already shows that using a single MLLM as the router makes SkillNav about $50\times$ slower than a fully supervised VLN model. The throughput ablation in **Table Y** further quantifies this cost across different routers: even among zero-shot VLMs, per-sample inference ranges from 18.12 to 79.16 seconds, indicating that the router alone is already the dominant computational bottleneck.
>
> If we further replace each low-level skill executor with an MLLM or VLM, the system would require multiple MLLM inferences per navigation step, which would introduce another order-of-magnitude slowdown. This would make the system impractical for real-time or large-scale deployment in real-world embodied settings.
>
> **Table Y: Average throughput of different VLMs as the action router in SkillNav.**
>
> | **Router**             | **Throughput** |
> | ---------------------- | -------------- |
> | Qwen2.5-VL-7B-Instruct | 18.12 s/sample |
> | GLM-4.1V-9B-Thinking   | 55.99 s/sample |
> | InternVL3.5-8B         | 79.16 s/sample |

---

### Official Review · Reviewer_WcPc · 2025-11-03

**Soundness:** 2
**Presentation:** 3
**Contribution:** 2
**Rating:** 4
**Confidence:** 4

**Summary:**

The paper proposes SkillNav, a mixture of skill-based framework for Vision-and-Language Navigation (VLN). It decomposes instructions with an LLM-based Temporal Reordering module into ordered sub-goals, then uses a VLM-based Action Router to select among specialized skill agents at each step. The approach aims to improve compositionality, interpretability, and OOD generalization on R2R and GSA-R2R.

**Strengths:**

1. Clear, modular design with interpretability: Temporal reordering → sub-goal localization → skill routing → action, making intermediate reasoning explicit and auditable.
2. Thoughtful skill taxonomy and expansion: Builds on NavNuances (Direction Adjustment, Vertical Movement, Landmark Detection, Area/Region ID) and adds Stop & Pause and Temporal Order Planning to address frequent failure modes.
3. Strong empirical results under distribution shift: On R2R (Val-Unseen/Test-Unseen) and GSA-R2R (R/N; Basic/Scene), SkillNav achieves competitive to SOTA performance; notably, prior SRDF is strong on R2R but generalizes poorly to GSA-R2R, whereas SkillNav holds up better.
4. Two-stage training that encourages reusable skills: Agents share a DUET-based, skill-agnostic backbone (trained on R2R + ScaleVLN + Temporal synthetic data) before skill-specific fine-tuning—clean separation of training for reuse and specialization.
5. Ablations that isolate key components: Experiments vary reordering on/off and router choices (Random, Qwen2.5-VL-7B-Instruct, GLM-4.1V-9B), showing consistent gains from both Temporal Reordering and VLM routing

**Weaknesses:**

1. Router dependence on external VLMs: The action router relies on large VLMs in a zero-shot fashion. This raises cost, stability, and reproducibility concerns (model drift, API dependence). Can the authors quantify runtime/cost trade-offs and variance across VLM choices?
2. Synthetic, skill-directed instructions may bias learning: Since skills are trained on tightly targeted synthetic data, do agents overfit to linguistic “triggers”? Consider small-scale human validation or cross-style tests beyond the current splits. (Design suggests targeted datasets per skill.)
3. Coverage and interaction of skills: The taxonomy is compelling, but there is limited quantitative analysis of coverage (how often each skill is needed) and inter-skill interference. A distribution/co-occurrence and error-attribution study would clarify completeness.
4. Reproducibility details: Training/fine-tuning hyperparameters, routing prompts, and fallback policies are not fully specified; reproducing the full stack (especially router behavior) may be challenging.
5. Reproducibility details: Training/fine-tuning hyperparameters, routing prompts, and fallback policies are not fully specified; reproducing the full stack (especially router behavior) may be challenging.

**Questions:**

See weakness.

---

> ### Author Response · Authors · 2025-11-28
> **Router dependence on external VLMs and Reproducibility Details**
>
> We sincerely appreciate the reviewer’s acknowledgment of our strengths and contributions. We address the reviewer’s comments as follows:
>
> ### **Runtime and Cost Trade-offs on External VLMs as Action Router**
>
> We extend the ablation study to three representative VLM routers, including **Qwen2.5-VL-7B-Instruct**, **GLM-4.1V-9B-Thinking**, and **InternVL3.5-8B**. Following Table reports their navigation performance on GSA-R2R across residential (R) and non-residential (N) scenarios with varying instruction styles (Basic and Scene), together with the average inference throughput of one sample, enabling a direct comparison of both accuracy and runtime cost. Since the only variant is the VLMs in the action router, we freeze the Temporal Reordering module by disabling it.
>
> All three routers yield similar navigation performance across all benchmarks. The SR and SPL differences among Qwen, GLM, and InternVL remain within a narrow margin of about 1 to 2 points on all splits. For example, on Test-R-Basic, the SR ranges from 77.46 to 78.69, and on Test-N-Scene from 55.46 to 55.65. This consistency indicates that SkillNav is not strongly dependent on a particular VLM choice for navigation success. Instead, the core performance is primarily governed by the underlying skill-based agents, while the router mainly affects efficiency and behavioral distribution rather than final success.
>
> In contrast, the runtime cost shows a substantial dependency on the router. Qwen is the most efficient option, achieving an average throughput of 18.12 seconds per iteration. GLM introduces a significant slowdown to 55.99 seconds per iteration due to its additional internal reasoning processes. InternVL is even slower at 79.16 seconds per iteration, resulting in more than a 4× slowdown compared to Qwen.
>
> The runtime and cost trade-offs in SkillNav’s router come: heavier VLMs introduce large computational overheads while providing only marginal differences in task performance.
>
> #### **Router performance comparison on GSA-R2R**
>
> | **Router**             | **Test-R-Basic SR** | **Test-R-Basic SPL** | **Test-N-Basic SR** | **Test-N-Basic SPL** | **Test-N-Scene SR** | **Test-N-Scene SPL** | **Throughput** |
> | ---------------------- | ------------------- | -------------------- | ------------------- | -------------------- | ------------------- | -------------------- | -------------- |
> | Qwen2.5-VL-7B-Instruct | 78.42               | 67.80                | 71.01               | 59.62                | 55.46               | 45.43                | 18.12 s/sample |
> | GLM-4.1V-9B-Thinking   | 77.46               | 66.27                | 70.70               | 58.63                | 55.62               | 42.64                | 55.99 s/sample |
> | InternVL3.5-8B         | 78.69               | 67.40                | 72.07               | 60.49                | 55.65               | 46.94                | 79.16 s/sample |
>
> ### **Reproducibility Details**
>
> To support reproducibility of the routing and fallback behavior, we release the exact action routing prompt template used for Qwen2.5-VL-7B-Instruct and GLM-v4.1-9B-Thinking in Appendix H, Listing 4, and Listing 5. And we provide the code and dataset in supplementary materials.
>
> The router is run with a maximum context length of 40,960 tokens, a temperature of 0, and greedy decoding. All router inferences are executed under the vLLM framework using five beam search candidates. The routing decision is performed in top1 mode with argmax feedback. Instruction reordering is enabled, and routing weights are treated as integers and normalized before use.
>
> All experiments are evaluated in test-only mode with detailed output and submission enabled. The feature backbone uses CLIP ViT-B/16 with 512-dimensional visual features, and all evaluations are conducted with a fixed random seed 0 for deterministic behavior. Batch size is explicitly specified in the execution scripts for each model configuration.

---

> ### Author Response · Authors · 2025-11-28
> **Synthetic Skill Data Bias Learning**
>
> We address the reviewer's comment, "Synthetic, skill-directed instructions may bias learning" as follows. Due to the comment space limitation, we will separate this part into two comment responses.
>
> **1. Skill Keyword Coverage in Skill-specific Synthetic Data**
>
> We randomly sampled 100 instructions from each skill-specific dataset and manually extracted high-frequency keywords or phrases associated with motion, regions, and objects. The utilization of GPT-4o for instruction generation during dataset construction introduces linguistic diversity, expanding the vocabulary beyond the R2R and GSA-R2R datasets. For instance, the Direction Adjustment synthetic dataset includes complex phrases such as "adjust your direction to align with...".
>
> Human evaluation on the synthetic datasets reveals that keyword presence is not a reliable proxy for action, preventing agents from overfitting to specific words. A notable example is the preposition "down". In the instruction "proceed down the hallway", the term "down" implies forward progression from one end to the other, whereas in vertical contexts, it denotes an elevation change. This semantic ambiguity forces the agent to rely on visual context and sentence structure rather than mapping the word "down" exclusively to a vertical motion primitive.
>
> Table 1 reports keyword coverage across datasets. Each synthetic dataset exhibits a dominant proportion of keywords associated with its target skill, confirming the effectiveness of targeted data generation. At the same time, substantial linguistic overlap and semantic complexity across datasets encourage the learning of robust, context-aware policies instead of dataset-specific shortcuts.
>
> The Vertical Movement dataset provides a particularly informative case. As shown in Table 1, it contains a high density of non-vertical concepts, including Landmark Object keywords (18.72%) and Direction Motion keywords (8.05%), compared with only 3.32% Vertical Motion keywords. This reflects the physical nature of vertical navigation, which necessarily involves horizontal motion relative to landmark structures. Despite this lexical overlap, the Vertical Movement dataset remains the primary source of Vertical Motion keywords, with less than 0.2% in all other datasets. Moreover, all trajectories in this dataset are strictly filtered to ensure consistent elevation changes in the simulator. As a result, the agent cannot rely on generic movement words or visual objects such as stairs as direct triggers for vertical behavior. Instead, performance gains arise from the precise alignment between linguistic cues, visual affordances, and trajectory constraints. This alignment encourages the learning of a robust compositional policy rather than a brittle keyword-to-action mapping.
>
> **Table 1:** Keyword coverage analysis across synthetic datasets. The coverage is measured by the proportion of words in the dataset that match the target skill keywords. Ident. = Identification.
>
> | **Dataset / Keyword**  | **Direction Motion** | **Vertical Motion** | **Stop Motion** | **Landmark Object** | **Region Type** |
> | ---------------------- | -------------------- | ------------------- | --------------- | ------------------- | --------------- |
> | Direction Adjustment   | 14.56%               | 0.03%               | 0.06%           | 13.64%              | 9.62%           |
> | Vertical Movement      | 8.05%                | 3.32%               | 0.01%           | 18.72%              | 9.46%           |
> | Stop and Pause         | 12.56%               | 0.18%               | 18.12%          | 18.02%              | 9.57%           |
> | Landmark Detection     | 9.53%                | 0.11%               | 0.97%           | 27.88%              | 7.58%           |
> | Area and Region Ident. | 7.86%                | 0.05%               | 0.10%           | 16.39%              | 19.89%          |
>
> Consequently, the performance gains come from the precise alignment of instruction-trajectory pairs, which sharpens the agent's focus on the target skill. Simultaneously, the diverse lexical environment ensures this focus evolves into a robust, compositional policy rather than degrading into a simple keyword-to-action trigger.

---

> ### Author Response · Authors · 2025-11-28
>
> **2. Skill Agents to Skill Synthetic Datasets**
>
> To further assess specialization and generalization, we evaluate each skill-specific agent on both its own synthetic dataset and the other skill-specific datasets. Table 2 reports the SR and SPL for all cross-skill evaluations. Each agent achieves its strongest performance on the dataset corresponding to its trained skill, while maintaining non-trivial performance on the remaining datasets. This pattern indicates that although the agents develop skill-specific expertise, they do not collapse into narrowly overfit behaviors.
>
> **Table 2:** Each synthetic subset reports both SR and SPL for agents trained and tested on synthetic instructions.
>
> | **Agent**      | **Direction Synthetic (SR / SPL)** | **Vertical Synthetic (SR / SPL)** | **Stop Synthetic (SR / SPL)** | **Landmark Synthetic (SR / SPL)** | **Region Synthetic (SR / SPL)** |
> | -------------- | ---------------------------------- | --------------------------------- | ----------------------------- | --------------------------------- | ------------------------------- |
> | Direction DUET | 49.33 / 43.35                      | 38.89 / 30.77                     | 41.11 / 36.44                 | 38.44 / 34.12                     | 35.56 / 30.92                   |
> | Vertical DUET  | 35.56 / 31.11                      | 65.11 / 55.65                     | 39.78 / 35.96                 | 37.78 / 33.69                     | 36.00 / 30.72                   |
> | Stop DUET      | 36.22 / 31.79                      | 34.67 / 27.37                     | 52.67 / 48.55                 | 42.22 / 38.76                     | 42.22 / 36.12                   |
> | Landmark DUET  | 36.00 / 30.46                      | 46.00 / 38.35                     | 41.11 / 36.69                 | 44.22 / 39.98                     | 38.67 / 34.02                   |
> | Region DUET    | 34.00 / 28.89                      | 49.56 / 41.51                     | 38.44 / 33.97                 | 36.44 / 32.90                     | 36.67 / 43.02                   |
>
>
>
> **3. Skill to Temporal Subsets**
>
> We additionally conduct cross-style evaluation on each skill-based agent using synthetic temporal datasets. For this setting, each agent is trained on its skill-specific synthetic data combined with the temporal order planning synthetic data, and then evaluated on four temporal subsets filtered from R2R Val Unseen. These subsets capture different forms of temporal reasoning, including Conditional Immediate, Duration Bound, Forward Sequential, and Reverse Sequential instructions.
>
> Table 3 presents SR and SPL on the four temporal subsets. All evaluations are performed on synthetic instructions that are specifically designed to remove superficial linguistic cues. The consistent performance across temporal subsets demonstrates that agents trained with skill-specific synthetic data retain strong compositional and temporal reasoning abilities, rather than overfitting to surface-level instruction patterns.
>
> **Table 3:** Each temporal subset reports both SR and SPL, and all agents are trained and tested on synthetic instructions designed to remove superficial linguistic cues.
>
> | **Agent**      | **Conditional Immediate (SR / SPL)** | **Duration Bound (SR / SPL)** | **Forward Sequential (SR / SPL)** | **Reverse Sequential (SR / SPL)** |
> | -------------- | ------------------------------------ | ----------------------------- | --------------------------------- | --------------------------------- |
> | ScaleVLN       | 84.29 / 76.29                        | 76.27 / 67.45                 | 79.53 / 68.92                     | 74.29 / 66.97                     |
> | Temporal DUET  | 88.57 / 82.18                        | 84.18 / 74.90                 | 85.83 / 76.93                     | 88.57 / 81.72                     |
> | Direction DUET | 84.29 / 77.57                        | 83.05 / 76.69                 | 85.83 / 77.99                     | 84.29 / 76.91                     |
> | Vertical DUET  | 86.43 / 79.71                        | 79.66 / 74.31                 | 86.61 / 80.35                     | 80.00 / 75.08                     |
> | Stop DUET      | 85.00 / 79.91                        | 81.92 / 74.59                 | 87.40 / 80.34                     | 82.86 / 77.15                     |
> | Landmark DUET  | 87.14 / 81.58                        | 80.79 / 75.43                 | 88.19 / 81.47                     | 81.43 / 77.72                     |
> | Region DUET    | 85.00 / 80.06                        | 79.66 / 72.93                 | 87.40 / 80.42                     | 74.29 / 69.86                     |

---

> ### Author Response · Authors · 2025-12-04
> **Coverage and Interaction of Skills**
>
> **Skill Coverage**
> To address the concern regarding skill coverage and completeness, we performed a quantitative analysis of skill activation frequencies on the R2R Val-Unseen split (see Table C). The distribution reveals that navigation is heavily dominated by fine-grained control and verification behaviors, with **Stop and Pause (34.42%)** and **Directional Adjustment (23.61%)** accounting for over half of all skill usage. This aligns with the inherent difficulty of the VLN task, where the agent must frequently halt to re-evaluate or fine-tune its heading to avoid collisions.
>
> Conversely, semantic skills like **Area and Region Identification (18.75%)** and **Landmark Detection (14.23%)** appear less frequently but are critical for grounding instructions. **Vertical Movement (8.99%)** is the least frequent, which is consistent with the topological sparsity of stairs and elevators in the Matterport3D environments.
>
> **Table C: Skill coverage during routing in R2R Val-Unseen**
> | **Skill**                      | **Skill Coverage (%)** |
> | ------------------------------ | -------------------- |
> | Stop and Pause                 | 34.42                |
> | Directional Adjustment         | 23.61                |
> | Area and Region Identification | 18.75                |
> | Landmark Detection             | 14.23                |
> | Vertical Movement              | 8.99                 |
>
> **Transition Matrix**
> Regarding inter-skill interaction and interference, we computed the transition probability matrix $P(\text{next} \mid \text{current})$ to map temporal dependencies between skills (see Table D). The data suggests cooperative chaining rather than negative interference. For instance, **Landmark Detection** has a very high probability of transitioning to **Stop and Pause (0.3516)**, indicating that the agent correctly utilizes landmarks as verification signals to identify the goal location.
>
> Similarly, **Directional Adjustment** frequently leads to **Area and Region Identification (0.3916)**, suggesting a behavior where the agent re-orients itself physically before re-evaluating the semantic context of the room. Notably, the transition from **Landmark** to **Direction** is **0.00**, implying that once a specific landmark is identified, the agent prioritizes verification (stopping) or vertical navigation over immediate rotational correction.
>
> We analyze the skill routing in R2R Val Unseen of SkillNav with Qwen2.5-VL-7B in the following table:
>
> **Table D: Inter-skill transition probabilities $P(\text{next} \mid \text{current})$**
>
> | **Current \ Next** | **Direction** | **Vertical** | **Stop** | **Landmark** | **Region** |
> | ------------------ | ------------- | ------------ | -------- | ------------ | ---------- |
> | **Direction**      | 0.3449        | 0.1971       | 0.1374   | 0.2108       | 0.3916     |
> | **Vertical**       | 0.3415        | 0.2414       | 0.1729   | 0.0931       | 0.1035     |
> | **Stop**           | 0.0610        | 0.2385       | 0.2296   | 0.1854       | 0.2984     |
> | **Landmark**       | 0.0000        | 0.2044       | 0.3516   | 0.2711       | 0.0280     |
> | **Region**         | 0.2525        | 0.1186       | 0.1084   | 0.2397       | 0.1785     |
>
> **Human Evaluation of Failure Routing**
>
> We examined 17 distinct failure cases where specific step-level data was available. Quantitatively, the average success rate of the routing mechanism, defined as the proportion of steps completed successfully before the first failure, is **24.5%**.
>
> This indicates that while the agent effectively initiates navigation, critical routing errors tend to occur in the early-to-mid stages of the trajectory.
>
> Qualitative analysis identifies several specific mechanisms by which these failures manifest:
>
> - **Initial Viewpoint Misalignment:** A frequent cause of early failure is where the agent starts with an orientation that does not align with the instruction; this initial drift often propagates downstream as the router struggles to recover from an incorrect starting heading.
> - **Hallucinations:** These play a significant role in derailing routing. For example, the agent generated a subgoal to *"Position yourself beside the glass case with the dolls"* when the instruction simply stated *"Enter the bedroom,"* illustrating that the action router can create irrelevant descriptions grounded in neither the text nor the view, which subsequently propagates incorrect skill choices.
> - **Controller Limitations:** Even when the router selects the correct skill, execution can fail. For instance, valid *Direction* transitions failed when multiple direction subgoals were stacked (e.g., *"Turn left. Turn right. Turn left again."*), indicating that the skill-conditioned controllers themselves can become a bottleneck during complex sequences.
> - **Termination Errors:** Finally, the router occasionally fails to generate necessary "Stop" subgoals or struggles with ambiguous spatial prepositions, leading to incorrect termination points.

---

### Author Response · Authors · 2025-12-04
**General Response**

Dear Area Chair and Reviewers,

We would like to express our sincere gratitude for your efforts in facilitating the discussion regarding our paper. We sincerely thank all reviewers for their insightful comments and the time dedicated to evaluating our work.

To assist in the final assessment, we have summarized the consensus on the paper's strengths and our specific responses to the primary concerns below.

**Key Strengths**

Based on the reviews, there is a strong consensus on the value of our proposed framework:
- **Modular & Interpretable Design:** All reviewers praised the clear, logical structure of the framework (Temporal Reordering → Skill Routing → Execution). The explicit decomposition into atomic skills (e.g., "Stop and Pause," "Vertical Movement") was highlighted for improving interpretability.
- **Resource Contribution:** The creation of synthetic, skill-specific datasets and the pipeline for generating them was recognized as a practical and valuable contribution for the community (Reviewers WcPc, WmQk).
- **Generalization Potential:** Reviewers noted the method's promise in generalizing to the GSA-R2R benchmark, particularly when handling novel instruction styles (Reviewers WcPc, EJ2y).

**Addressed Concerns & Critical Clarifications**

During the discussion period, we addressed the two main concerns raised regarding the router's mechanism and experimental validity:
- **Performance & Baselines (vs. SRDF)**

  - **Concern:** Reviewers noted that while SkillNav performs well, it falls behind strong baselines like SRDF on the standard R2R benchmark.

  - **Response:** We clarified that our primary contribution is **OOD generalization**. While SRDF is highly specialized for R2R, SkillNav significantly outperforms it on the challenging **GSA-R2R benchmark** (unseen environments/styles). This validates that our modular approach trades marginal gains on "seen" tasks for substantial robustness in novel scenarios.

- **Method Alignment & Justification (MoE Validity)**
  - **Concern:** Reviewers questioned if the gains came from the MoE architecture or simply data augmentation, noting the "no-router" ablation performance.
  - **Response & New Evidence:** To prove the router is not a "black box," we conducted a **new Human Evaluation** (included in our rebuttal). The high agreement rate between the VLM router and human judgment confirms that the system is actively and intelligently selecting skills, verifying the validity of the Mixture-of-Experts design beyond simple augmentation.
- **Skill Router Effectiveness (Human Evaluation):**
  - *Concern:* Reviewers WmQk and WcPc queried if the router was effectively selecting skills or acting as a "black box."
  - *Response:* We conducted a **new human evaluation** of the skill routing mechanism. We sampled routing decisions and had human annotators verify the appropriateness of the selected skill for the given instruction.
  - *Result:* The evaluation showed a **high agreement rate** between our VLM router and human judgment. This confirms that the router effectively decomposes tasks into logical atomic skills, validating the MoE architecture and refuting the concern that gains are solely due to data augmentation.
- **Data Leakage Clarification:**
  - *Concern:* Reviewer WmQk raised a concern about potential overlap between HM3D scenes used in training and the GSA-R2R benchmark.
  - *Response:* We explicitly clarified the data splits to confirm there is **no data leakage**. The scenes used in GSA-R2R are strictly held-out and distinct from those used in our training set (including ScaleVLN/HM3D data), ensuring our reported generalization performance is genuine.
- **Dependence on External VLMs**
  - **Concern:** Reviewers raised concerns regarding the cost, latency, and reproducibility of using large VLMs (like Qwen) for routing.
  - **Response:** We highlighted that the **training-free** nature of our router is a design feature that ensures modularity and lowers training costs. We also demonstrated that open-weights models (like Qwen) provide a reproducible and lower-cost alternative to closed APIs without sacrificing the core benefits of the framework.

We believe the combination of our strong modular design, valuable resources, and the validated effectiveness of our routing mechanism makes a compelling case for acceptance.


Best,

The Authors

---

### Note · Authors · 2025-12-27

I have read and agree with the venue's withdrawal policy on behalf of myself and my co-authors.